# Mapping immune variation and *var* gene switching in naive hosts infected with *Plasmodium falciparum*

**Kathryn Milne[1†], Alasdair Ivens[1,2†], Adam J Reid[3†], Magda E Lotkowska[3], Aine O'Toole[2,4], Geetha Sankaranarayanan[3], Diana Munoz Sandoval[1,5], Wiebke Nahrendorf[1], Clement Regnault[6,7], Nick J Edwards[8], Sarah E Silk[8], Ruth O Payne[8], Angela M Minassian[8], Navin Venkatraman[8], Mandy J Sanders[3], Adrian VS Hill[8], Michael Barrett[6,7], Matthew Berriman[3], Simon J Draper[8], J Alexandra Rowe[1,2‡*], Philip J Spence[1,2‡*]**

[1]Institute of Immunology and Infection Research, University of Edinburgh, Edinburgh, United Kingdom; [2]Centre for Immunity, Infection and Evolution, University of Edinburgh, Edinburgh, United Kingdom; [3]Wellcome Sanger Institute, Cambridge, United Kingdom; [4]Institute of Evolutionary Biology, University of Edinburgh, Edinburgh, United Kingdom; [5]Instituto de Microbiologia, Universidad San Francisco de Quito, Quito, Ecuador; [6]Wellcome Centre for Integrative Parasitology, University of Glasgow, Glasgow, United Kingdom; [7]Glasgow Polyomics, University of Glasgow, Glasgow, United Kingdom; [8]The Jenner Institute, University of Oxford, Oxford, United Kingdom

**\*For correspondence:**
Alex.Rowe@ed.ac.uk (JAR);
Philip.Spence@ed.ac.uk (PJS)

[†]These authors contributed equally to this work
[‡]These authors also contributed equally to this work

**Competing interests:** The authors declare that no competing interests exist.

**Abstract** Falciparum malaria is clinically heterogeneous and the relative contribution of parasite and host in shaping disease severity remains unclear. We explored the interaction between inflammation and parasite variant surface antigen (VSA) expression, asking whether this relationship underpins the variation observed in controlled human malaria infection (CHMI). We uncovered marked heterogeneity in the host response to blood challenge; some volunteers remained quiescent, others triggered interferon-stimulated inflammation and some showed transcriptional evidence of myeloid cell suppression. Significantly, only inflammatory volunteers experienced hallmark symptoms of malaria. When we tracked temporal changes in parasite VSA expression to ask whether variants associated with severe disease rapidly expand in naive hosts, we found no transcriptional evidence to support this hypothesis. These data indicate that parasite variants that dominate severe malaria do not have an intrinsic growth or survival advantage; instead, they presumably rely upon infection-induced changes in their within-host environment for selection.

## Introduction

There is enormous diversity in the human response to identical immune challenge. This variation is currently being interrogated in large cohorts of healthy volunteers to pinpoint the genetic and non-genetic factors that dictate human immune decision-making (*Bakker et al., 2018*; *Brodin et al., 2015*; *Li et al., 2016*; *Patin et al., 2018*; *Piasecka et al., 2018*; *Ter Horst et al., 2016*). Controlled human malaria infection (CHMI) provides a well-established challenge model in which to examine immune variation in vivo, and heterogeneity in the host response to *Plasmodium falciparum* is well-recognised (*Roestenberg et al., 2012*; *Yap et al., 2020*). Prior work suggests a dichotomy in immune decision-making with some individuals triggering an inflammatory response, which is characterised by high circulating levels of IFNγ, whilst others mount an alternative response distinguished

by the early release of TGFβ and suppression of IFNγ signalling (*de Jong et al., 2020*; *Harpaz et al., 1992*; *Walther et al., 2005*; *Walther et al., 2006*). The precise mechanisms underpinning these divergent responses remain unknown.

Genome-wide transcriptional profiling has the potential to map immune decision-making in unprecedented detail and previous CHMI studies have revealed an upregulation of genes associated with pathogen detection, NFκB activation, and IFNγ signalling consistent with a systemic inflammatory response (*Ockenhouse et al., 2006*; *Tran et al., 2019*; *Tran et al., 2016*). A transcriptional signature of an alternative immune response has not been described in naive hosts. Nevertheless, these prior studies were primarily designed to measure shared or common signatures of infection, potentially masking variation between volunteers. Only one study (*Burel et al., 2017*) has specifically investigated transcriptional variation in CHMI and it reported two distinct outcomes based on host microRNA profiles; notably, the authors were able to correlate the induction of three specific miRNAs with a lower parasite burden suggesting that events early in infection may have important downstream consequences. Identifying the sources of immune variation may therefore reveal some of the factors that underpin the clinical heterogeneity of falciparum malaria. In this context, direct intravenous challenge with blood-stage parasites provides a unique opportunity to assess the contribution of host-intrinsic mechanisms of variation as it ensures all volunteers receive an identical immune challenge (*Duncan and Draper, 2012*).

The potential role of parasite factors in influencing immune variation has not yet been explored in CHMI. Expression of a subset of parasite variant surface antigens (VSA) known as group A and DC8 *var* genes is associated with severe malaria in both children and adults (*Duffy et al., 2019*; *Jespersen et al., 2016*; *Kyriacou et al., 2006*; *Lavstsen et al., 2012*; *Tonkin-Hill et al., 2018*; *Warimwe et al., 2012*). These virulence-associated genes encode PfEMP1 adhesion molecules that mediate sequestration of infected red cells in the microvasculature. Group A/DC8-expressing parasites are thought to be more pathogenic than those expressing other *var* types (groups B and C) and can bind the endothelium with high affinity in sensitive sites, such as brain (*Jensen et al., 2020*). It is unknown whether expression of these variants can influence the immune response, for example by causing dysregulated activation that leads to widespread collateral tissue damage. Or conversely, whether inflammation might preferentially support the survival and expansion of these variants by upregulating their binding sites on the endothelium.

It has been observed in previous CHMI studies that parasites transcribe a broad array of *var* genes in naive hosts, with predominant expression of group B variants and no marked differences between volunteers (*Bachmann et al., 2019*; *Bachmann et al., 2016*). Indeed, we have shown that mosquitoes reset malaria parasites to ensure diverse expression of their VSA repertoire at the start of the blood cycle (*Spence et al., 2013*). It therefore remains unclear how group A and DC8 *var* genes come to dominate the PfEMP1 landscape in severe malaria. Antibody-mediated clearance of parasites expressing group B variants is a possible explanation, but is unlikely because severe malaria develops in hosts with low or absent immunity during their first few infections of life (*Gonçalves et al., 2014*). It has instead been suggested that parasites expressing group A/DC8 *var* genes rapidly and preferentially expand because they have an intrinsic growth and/or survival advantage in naive hosts (*Abdi et al., 2017*; *Bull et al., 2000*; *Jensen et al., 2004*). This may be a consequence of enhanced cytoadherence (reducing their mechanical clearance in the spleen), rosetting (increasing their re-invasion efficiency) or specificity for tissues that are better able to support their maturation (decreasing cycle length). This explanation is widely accepted but is not based on substantial evidence.

In this study, we have used a blood challenge model to investigate host-intrinsic variation in the immune response to falciparum malaria and examined the interplay between parasite and host factors in shaping outcome of infection. This model has the advantage that an identical immune challenge is given to all volunteers and furthermore transcriptional profiling can be carried out on parasites at the start and end of infection to track changes in VSA expression through time. We set out to test the specific hypotheses that **a**, expression of group A and DC8 *var* genes would preferentially increase as the infection progressed and **b**, there would be a measurable relationship between inflammation and parasite VSA expression, which would shape the clinical outcome of infection.

## Results

### Immune variation in falciparum malaria

To explore host-intrinsic variation in the immune response to *P. falciparum*, a cohort of 14 malaria-naive volunteers were infected with an equal number of blood-stage parasites (clone 3D7) by direct intravenous inoculation (*Supplementary file 1*). This enabled the pathogenic blood cycle to be initiated in every volunteer within a 30-min window (*Payne et al., 2016*). Systemic changes in the host response were then captured through time by transcriptionally profiling whole blood every 48 hr from the day before infection until the day of diagnosis (the point of drug treatment). This time-point varied between volunteers, occurring 7.5–10.5 days post-infection when two out of three diagnostic criteria were met (positive thick blood smear and/or parasite density >500 parasites ml$^{-1}$ by qPCR and/or symptoms consistent with malaria). To reveal the diversity of responses within our cohort, each volunteer's time-course was analysed independently by tracking their dynamic changes in gene expression as the infection progressed. Importantly, this approach does not assume shared features between individuals, nor does it bias against uncommon (or rare) responses.

As a first step, we measured the variance of every gene through time and visualised the 100 protein-coding genes with highest variance in each volunteer (*Figure 1—source data 1*). To control for baseline variation in gene expression, uninfected control volunteers were also analysed. We found that variance in one third of infected volunteers (4/14) was comparable to the uninfected control group, suggesting these four volunteers did not respond to infection (*Supplementary file 2*). To explore this further, we merged the top 100 gene lists from all infected volunteers to produce a single non-redundant list of the most dynamically expressed protein-coding genes across the cohort (n = 517 unique genes, henceforth called the 517-gene superset); we could then directly compare the transcriptional response between individuals. Accordingly, we performed a principal component analysis (PCA) of these genes through time, and for each volunteer we determined the Euclidean distance travelled to quantify the magnitude of their response (*Figure 1—source data 1* and *Supplementary file 2*). Using the variance and distance travelled metrics, we then set two thresholds to identify volunteers that triggered a measurable response to blood challenge (see Materials and methods); four volunteers failed to cross both thresholds (v018, v012, v208, and v020) and were therefore grouped and labelled unresponsive. A pairwise comparison between the diagnosis and pre-infection samples obtained from these four volunteers confirmed that there were zero differentially expressed genes in this group (adj p<0.05). Immune quiescence is therefore a common outcome of controlled human malaria infection.

In contrast, variance (*Figure 1—source data 1*) and distance travelled (*Figure 1—figure supplement 1*) both suggested that the remaining 10 volunteers made a robust response to infection (*Supplementary file 2*). And when we clustered and visualised the 517-gene superset expression data we found that these volunteers could separate easily into two groups according to whether they up- or downregulated gene expression at diagnosis (*Figure 1* and *Figure 1—source data 2*). Based on this observation, we grouped the eight volunteers who upregulated gene clusters 2 and 4 and performed a pairwise comparison between their diagnosis and pre-infection samples; we uncovered 2028 differentially expressed genes (adj p<0.05) (*Supplementary file 3*). To gain insight into the biological functions of this response, we used ClueGO (*Bindea et al., 2009*; *Mlecnik et al., 2014*) to first identify the significant GO terms associated with these genes and then place them into a functionally organised non-redundant gene ontology network. ClueGO uncovered 217 GO terms across 34 functional groups (*Supplementary file 4*); the most significant groups related to interferon signalling, activation of myeloid cells, production of inflammatory molecules, and activation of T cells (*Figure 2a–b*). These terms all indicate systemic interferon-stimulated inflammation (a well-described host response to malaria in mouse and human studies [*Montes de Oca et al., 2016*; *Spaulding et al., 2016*; *Yu et al., 2016*]) and we therefore labeled this group inflammatory. We then biologically validated ClueGO's in silico finding by measuring the plasma concentrations of two interferon-stimulated chemokines, CXCL9 and CXCL10. These chemokines are primarily produced by myeloid cells and specifically function to recruit and activate T cells (*Groom and Luster, 2011*). We found a > 2-fold increase in one or both of these chemokines at diagnosis in every volunteer in this group but no other volunteer (*Figure 2c*).

The final two volunteers were also grouped (and given the suppressor tag) based on their apparent downregulation of clusters 1 and 3 at diagnosis (*Figure 1*). Significantly, a pairwise comparison

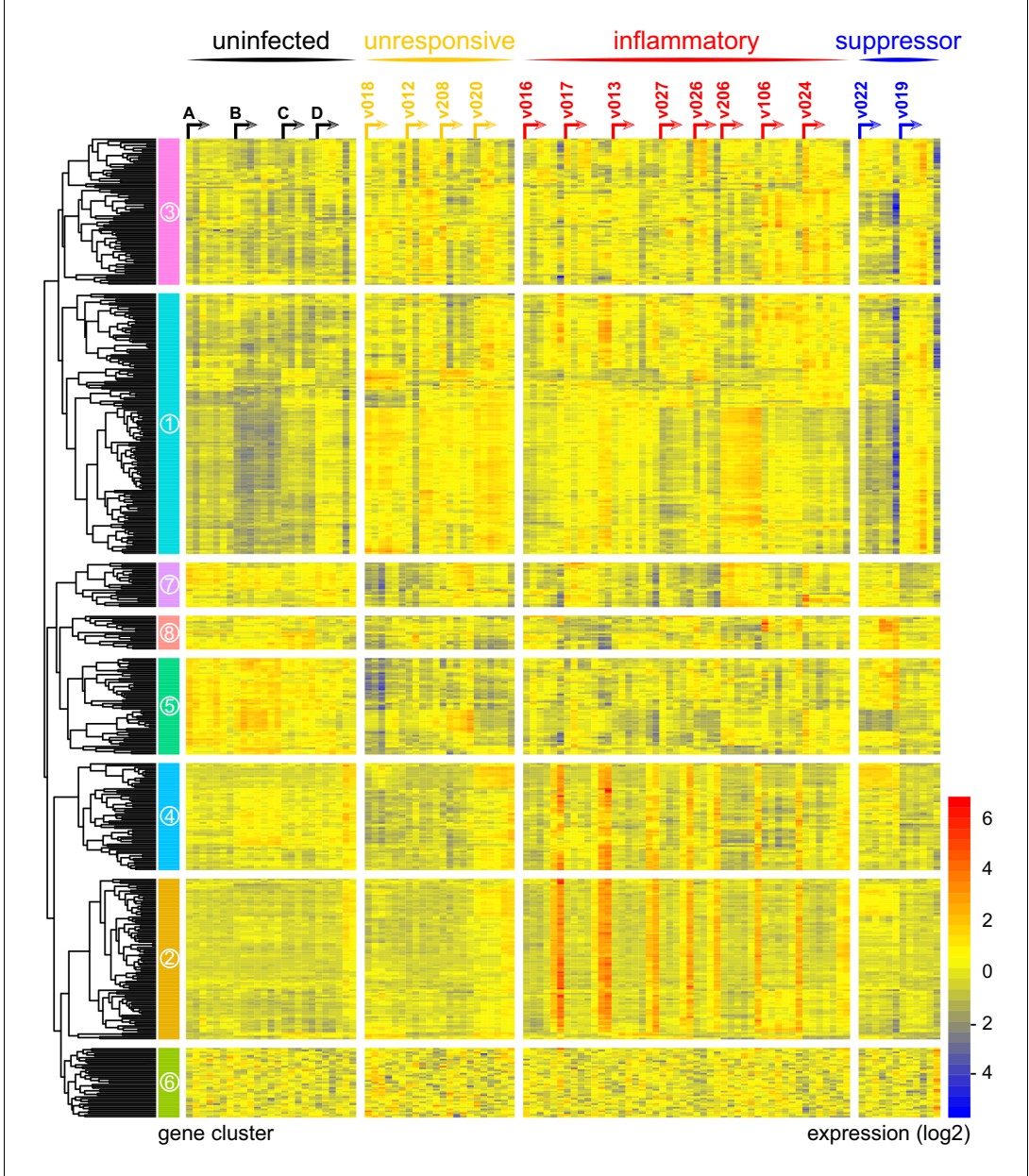

**Figure 1.** Immune variation in falciparum malaria. Log2 expression values of 517 protein-coding genes in whole blood during infection. Genes (rows) are ordered by hierarchical clustering whereas whole blood samples (columns) are ordered by volunteer and time-point (pre-infection to diagnosis, left to right). Arrows start from the pre-infection sample and volunteers are grouped by host response. Uninfected controls demonstrate minimal within-host variation in expression of these genes. Median sample number per volunteer = 6.

The online version of this article includes the following source data and figure supplement(s) for figure 1:

**Source data 1.** An identical immune challenge leads to diverse outcomes in falciparum malaria.

**Source data 2.** Log2 transformed expression values of the 517-gene superset in whole blood during infection.

**Figure supplement 1.** Immune quiescence is a common early outcome of infection.

between their diagnosis and pre-infection samples revealed a core signature of 77 genes that were differentially expressed in response to blood challenge (adj p<0.05, *Supplementary file 5*). The majority of these genes (62/77) were downregulated and crucially >85% were unique to this group, including key regulators of monocyte and neutrophil differentiation (*Csf3r*) and recruitment and activation (*Cxcl7*, *Cxcr2*, *Clec4e*, *Clec7a*, and *Mnda*, *Figure 2d*). A direct pairwise comparison between the inflammatory and suppressor groups confirmed that each of these six immune genes was

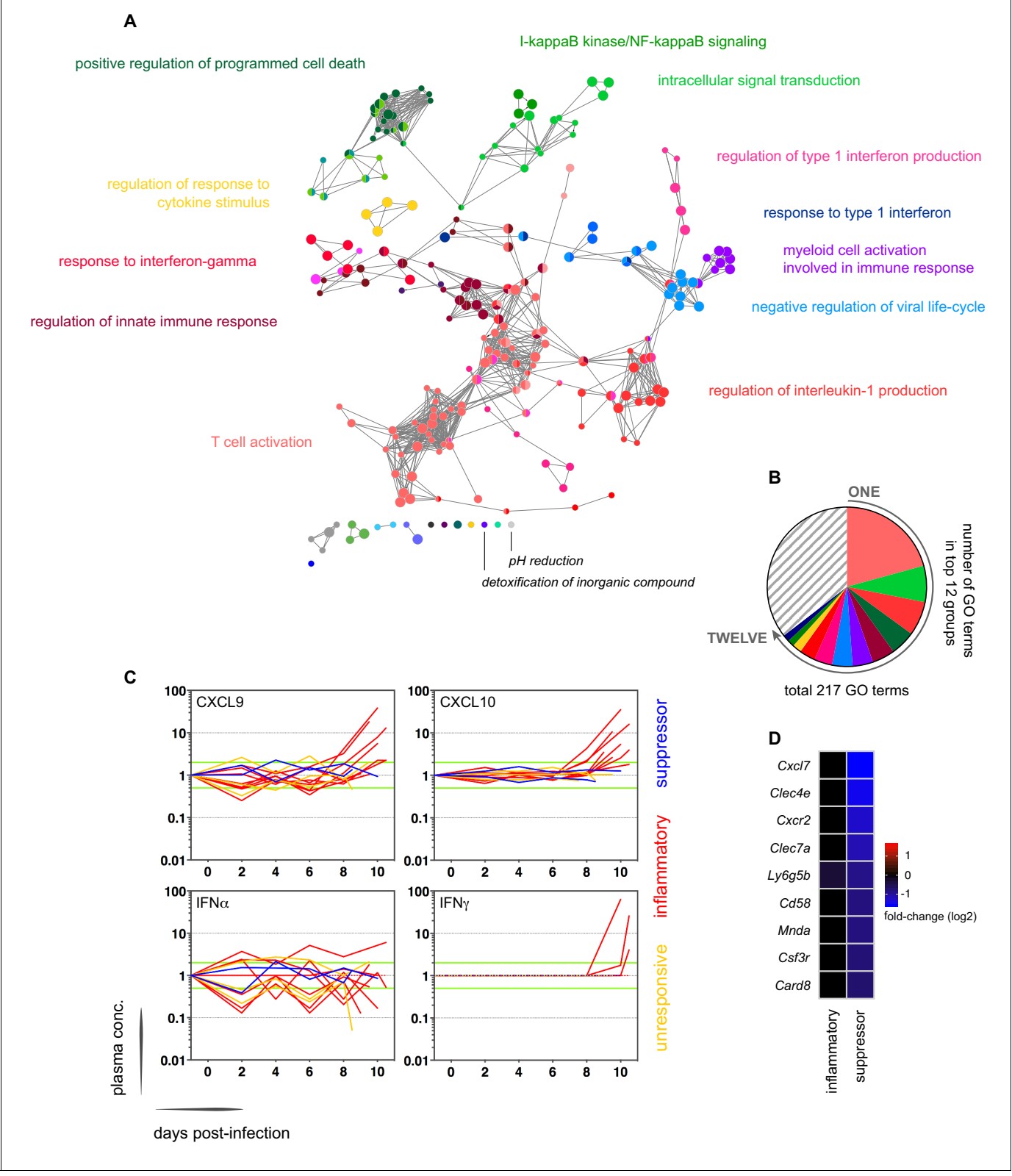

**Figure 2.** Interferon-stimulated inflammation is the dominant response to blood-stage infection. (**A**) Gene ontology network of 2028 genes differentially expressed at diagnosis in the inflammatory group. Each node represents a significantly enriched GO term (adj p<0.05) and node size is determined by significance (bigger nodes have lower p values). Nodes are interconnected according to their relatedness (kappa score >0.4) and grouped if they are connected and share >40% genes. Each functional group is then given a unique colour and the leading GO term in the top 12 groups is highlighted.

*Figure 2 continued on next page*

*Figure 2 continued*

Two GO terms of interest, which are not part of any functional group, are also shown in italics. (**B**) The proportion of GO terms in each of the top 12 functional groups; collectively, these account for two thirds of all significantly enriched GO terms in inflammatory volunteers. (**C**) Plasma concentration of interferon alpha and gamma and interferon-stimulated chemokines (CXCL9 and CXCL10) during infection. One line represents one volunteer (no data for v020) and lines are colour-coded by host response. For each volunteer, all data points are normalised to their own baseline (day −1); horizontal green lines represent a twofold increase or decrease compared to baseline. (**D**) Log2 fold-change of nine immune genes involved in myeloid cell differentiation and activation in whole blood at diagnosis. Data are presented relative to pre-infection samples and all genes are significantly downregulated in the two suppressor volunteers (adj p<0.05).

The online version of this article includes the following figure supplement(s) for figure 2:

**Figure supplement 1.** Sporozoites do not trigger a systemic transcriptional response in human malaria.

differentially expressed at diagnosis, together with a further 887 genes (adj p<0.05, *Supplementary file 6*). Evidently, these host response profiles do not overlap and instead likely represent distinct in vivo outcomes of immune decision-making. Nevertheless, a much larger sample size would be required to fully elucidate immune suppression as an alternative response to *P. falciparum*.

Collectively, these data reveal marked variation in the human immune response to falciparum malaria with at least three possible outcomes of blood-stage infection – immune quiescence, interferon-stimulated inflammation, and suppression of myeloid-associated gene expression.

## Sporozoites have a limited role in shaping the host response to blood-stage infection

We next explored the influence of the pre-erythrocytic-stages of infection on the immune response to blood-stage parasites. Whilst blood challenge uses a recently mosquito-transmitted parasite line (< 3 blood cycles from liver egress [*Cheng et al., 1997*]) this model nevertheless bypasses the skin and liver stages, which have been proposed to elicit regulatory mechanisms that could modify the subsequent immune response to blood-stage infection (*Guilbride et al., 2010*). We therefore inoculated five additional malaria-naive volunteers with the same parasite genotype but this time by the natural route of infection, mosquito bite (*Supplementary file 1*). As before, we transcriptionally profiled whole blood from the day before infection until diagnosis (day 11), including the final day of liver-stage infection (day 6). And once more, every volunteer was analysed independently this time using EdgeR to identify highly dispersed genes within each time-course (see Materials and methods). After merging these lists, we produced a single non-redundant list of the most dynamically expressed genes across the mosquito challenge cohort (n = 117 unique genes, henceforth called the 117-gene superset) (*Supplementary file 7*). When we clustered and visualised the expression data within this superset we found that 116 out of 117 genes were upregulated exclusively during the blood cycle (*Figure 2—figure supplement 1a–b*). And when we examined the biological functions of these genes ClueGO identified response to interferon gamma (GO: 0034341) as the most significantly enriched GO term (*Supplementary file 8*). Notably, this was also the most significant GO term in inflammatory volunteers infected by blood challenge (*Supplementary file 4*), indicating concordance between the two infection models. In agreement, 19 of the 21 genes that displayed the highest fold change in inflammatory volunteers (blood challenge) were also upregulated in the 117-gene superset (mosquito challenge, *Supplementary file 7*). And at protein level, high circulating levels of CXCL10 were detectable at diagnosis in four of the five volunteers infected by mosquito bite (*Figure 2—figure supplement 1c*). Our data therefore demonstrate that interferon-stimulated inflammation is the dominant human response to blood-stage parasites regardless of the route of infection. Furthermore, our data fail to provide compelling evidence of a systemic transcriptional response to liver-stage infection in human malaria (*Figure 2—figure supplement 1b*). Taken together, these findings indicate that the pre-erythrocytic-stages of infection may have a limited role in shaping the immune response during the pathogenic blood cycle.

## Systemic inflammation coincides with the onset of clinical symptoms

We therefore moved on to investigate the factors that influence human immune variation in the blood challenge model and the consequences of these divergent paths. In the first instance, we

asked whether the rate of parasite replication could shape the host response to infection; after all, a simple assumption might be that inflammation is triggered by fast-growing parasites. The parasite multiplication rate (PMR, modelled from the qPCR data [*Payne et al., 2016*]) varied considerably between volunteers (range 6.7–12.8) despite every individual receiving an equal number of infected red cells (*Supplementary file 1*). However, the parasite multiplication rate of the inflammatory group was not significantly different to that of unresponsive or suppressor hosts (*Figure 3a–b*) and therefore appears to have little bearing on the outcome of immune decision-making. Inflammatory volunteers did, however, tend to have a longer course of infection (*Figure 3a*) and hence the parasite burden at diagnosis was higher for the inflammatory group (median 30,309 parasites ml$^{-1}$, range 9133–273,247) than the rest of the cohort (median 7899 parasites ml$^{-1}$, range 1440–19,670) (Mann Whitney test, p=0.02). Parasite burden may therefore promote an inflammatory response in this challenge model but cannot fully explain it; for example, 2/4 unresponsive volunteers (v018 and v020) had parasite densities that were comparable to 4/8 inflammatory volunteers at diagnosis (v017, v027, v026, and v106, *Supplementary file 1*). Our data therefore support previous observations that the parasite density threshold required to trigger inflammation is highly variable between individuals (*Gatton and Cheng, 2002*; *Molineaux et al., 2002*).

We next set out to ask whether immune variation had clinical consequences. Symptomatology was scored throughout infection with adverse events graded as absent (0), mild (1), moderate (2), or severe (3). Volunteers in the unresponsive and suppressor groups were essentially asymptomatic at diagnosis, whereas inflammatory volunteers exhibited more frequent and/or severe adverse events, including fever (*Figure 3—figure supplement 1* and *Figure 3—figure supplement 2a*). When we then calculated a cumulative clinical score for each volunteer (by summing all adverse events throughout infection) we found a strong positive correlation between symptoms and the intensity of inflammation (*Figure 3c*). In contrast, there was no correlation between clinical score and parasite multiplication rate (*Figure 3d*). We then measured circulating levels of Angiopoietin-2, a biomarker of endothelium dysfunction and indicator of disease severity (*Yeo et al., 2008*); only one volunteer displayed elevated levels of Angiopoietin-2 (*Figure 3—figure supplement 2b*) and this individual was the most inflammatory host in the cohort (based on upregulation of plasma CXCL10 and IFNγ). What's more, haematological analysis revealed that only inflammatory volunteers experienced acute lymphopenia, a conserved hallmark of clinical malaria (*Figure 3—figure supplement 2c–d*; *Hviid et al., 1997*). Symptomatology is therefore not driven by the rate of parasite replication in controlled human malaria infection. Instead, clinical outcome seems to depend upon the intensity of inflammation, which varies widely between hosts.

In light of these findings, we examined the potential impact of inflammation on host metabolism to better resolve the consequences of immune variation. Here, we used an LC-MS-based metabolomics platform to measure changes in circulating metabolites during infection, and applied range-scaling to ensure all metabolites (and their temporal fluctuations) were equally weighted (*van den Berg et al., 2006*). When we compared the three most inflammatory and symptomatic volunteers to unresponsive and suppressor hosts we could find no metabolic signature of inflammation that was robust to multiple testing (zero metabolites with an FDR-corrected p value < 0.1). On the other hand, if we simply considered all volunteers as a single group and compared their post-infection time-points (day 8 and diagnosis) against pre-infection samples we could identify a conserved and persistent signature of blood-stage infection (*Figure 3—figure supplement 3*). This was characterised by increased adenosine (promotes vasodilation [*Ralevic and Dunn, 2015*]), choline depletion (supports parasite membrane synthesis [*Witola and Ben Mamoun, 2007*]) and reduced lysoPC (induces gametocytogenesis [*Brancucci et al., 2017*, *Figure 3e–f*]). Evidently, volunteers who are immune quiescent are nevertheless metabolically responsive to *P. falciparum*, and their response is indistinguishable from inflammatory hosts.

All together, these data reveal that immune decision-making, systemic metabolism and parasite replication all operate independently in the early stages of the blood cycle. And they further reveal a clear relationship between the intensity of inflammation and symptomatology in falciparum malaria.

## Parasite variants associated with severe disease do not rapidly expand in naive hosts

To investigate the origin of immune variation and gain further insight into the factors that can shape outcome of infection we measured parasite VSA expression. To this end, we applied ultra-low input

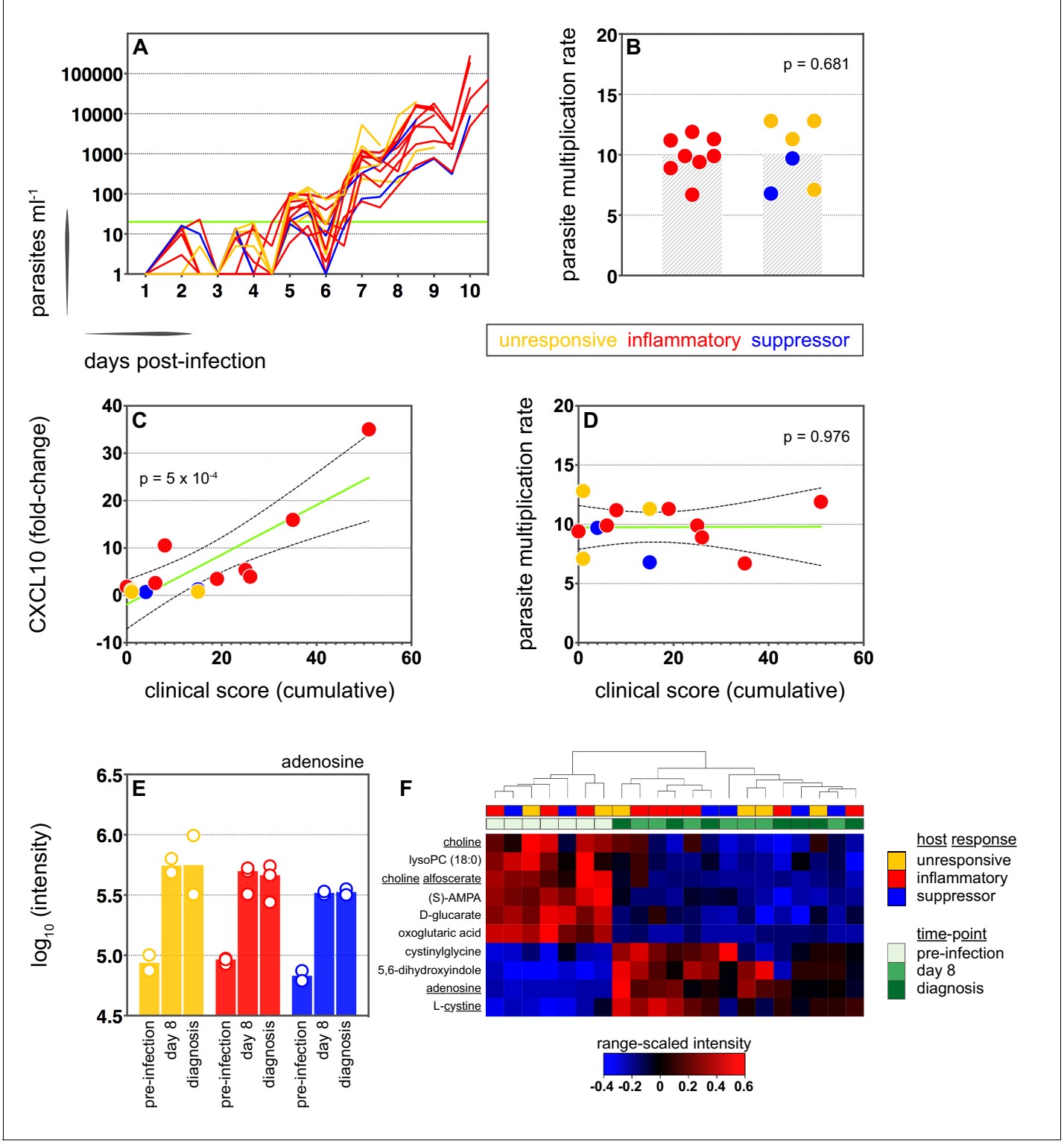

**Figure 3.** Systemic inflammation coincides with the onset of clinical symptoms. (**A**) Parasite growth curves colour-coded by host response; each line represents one volunteer. Blood samples were collected every 12 hr for qPCR analysis of circulating parasite density and the horizontal green line represents the lower limit of quantification (20 parasites ml$^{-1}$). (**B**) Parasite multiplication rates colour-coded by host response; each dot represents one volunteer and shaded areas show the mean value. A Mann Whitney test was used to ask whether the parasite multiplication rate observed in the inflammatory group was different to all other volunteers (p value is shown). (**C and D**) Linear regression of CXCL10 (**C**) or parasite multiplication rate (**D**) plotted against clinical score (the sum of adverse events during infection). CXCL10 fold-change measures plasma concentration at diagnosis over

*Figure 3 continued on next page*

*Figure 3 continued*

baseline (day −1). One dot represents one volunteer (no data for v020) and dots are colour-coded by host response. The green line represents the best-fit model (p value of the slope is shown) and dashed lines are the 95% confidence intervals. (E) Log10 transformed intensity values of adenosine in plasma during infection. An authentic standard was run in tandem with all samples to validate adenosine detection. (F) Range-scaled intensity values of 10 plasma metabolites that were differentially abundant during infection. Metabolites (rows) and samples (columns) are ordered by hierarchical clustering. Note that an authentic standard was used to validate detection of all underlined metabolites and the full name for oxoglutaric acid is 4-hydroxy-2-oxoglutaric acid. (E and F) Only plasma samples from the most inflammatory and symptomatic volunteers (v016, v017, and v013), the suppressor volunteers (v022 and v019) and two unresponsive volunteers (v018 and v208) were analysed for metabolite abundance.

The online version of this article includes the following figure supplement(s) for figure 3:

**Figure supplement 1.** Adverse events after blood challenge.

**Figure supplement 2.** Inflammation is linked to hallmark symptoms of malaria.

**Figure supplement 3.** A conserved early metabolic signature of falciparum malaria.

RNA-sequencing methodology to parasites isolated during CHMI and mapped their transcriptional profiles from the start to end of infection. In the first instance, we examined the hierarchy of *var* gene expression in the inoculum used for blood challenge; this derived from a single malaria-infected volunteer infected by mosquito bite and was cryopreserved ex vivo (without culture) after just three cycles of asexual replication (*Cheng et al., 1997*). Although each individual blood-stage parasite will only transcribe a single *var* gene (*Freitas-Junior et al., 2005*) the parasite population as a whole can express many different variants within a host. In agreement, we found more than 20 *var* genes highly transcribed in the inoculum; notably, the expression hierarchy was dominated by group B variants and there was very low (or undetectable) transcription of most virulence-associated group A or DC8 *var* genes (*Figure 4a* and *Figure 4—source data 1*). One exception was PF3D7_0400400 (previously PFD0020c), which encodes a group A brain endothelial cell binding PfEMP1 variant (*Claessens et al., 2012*). We further found that group B *var* genes dominated the PfEMP1 landscape when parasites were isolated from our mosquito challenge cohort, albeit with a higher proportion of *var* gene reads mapping to group A and with two DC8-like variants highly transcribed in some volunteers (PF3D7_0600200 (aka PFF0010w) and PF3D7_0800300 (aka PF08_0140)) (*Figure 4a*, *Figure 4—figure supplement 1a–b* and *Figure 4—source data 1*). Our data therefore confirmed previous observations that whilst malaria parasites exit the human liver expressing a diverse *var* gene repertoire, variants associated with severe disease constitute a minority population (*Bachmann et al., 2019*; *Bachmann et al., 2016*; *Peters et al., 2002*; *Wang et al., 2009*).

Current theory predicts that group A and DC8 variants should rapidly and preferentially expand in naive hosts because of an intrinsic growth and/or survival advantage favouring their selection (*Abdi et al., 2017*; *Bull et al., 2000*; *Jensen et al., 2004*). Blood challenge provides a unique opportunity to directly test this hypothesis as it increases the number of asexual replication cycles over which variant switching and selection can take place (cf. mosquito challenge [(*Draper et al., 2018*)]). Parasites were therefore isolated from volunteers infected by blood challenge and processed immediately (without culture) for RNA-sequencing; after six cycles of replication, the expression hierarchy of *var* genes was essentially unchanged from that observed in the inoculum (*Figure 4a* and *Figure 4—figure supplement 1b*). Indeed, the group B *var* gene PF3D7_1041300 remained the most highly expressed variant in every volunteer, and transcription of group A and DC8 *var* genes remained low or undetectable at diagnosis (*Figure 4—source data 1*). Surprisingly, this included the virulence-associated brain endothelial cell binding variant PF3D7_0400400, which was transcribed at the start of infection (*Figure 4a*). We could further highlight the significance of this result by looking only at intron-spanning reads, which must derive from functional *var* gene transcripts (rather than regulatory antisense transcripts, *Amit-Avraham et al., 2015*). We found that PF3D7_0400400 was functionally transcribed in the inoculum but intron-spanning reads were absent in all volunteer samples (*Figure 4—figure supplement 1c*). Differential gene expression analysis (using DESeq2) then confirmed that no *var* gene was more highly transcribed at the end of infection as compared to the start (adj p<0.01). Evidently, there was no preferential expansion of group A or DC8 variants within the time-frame of this study.

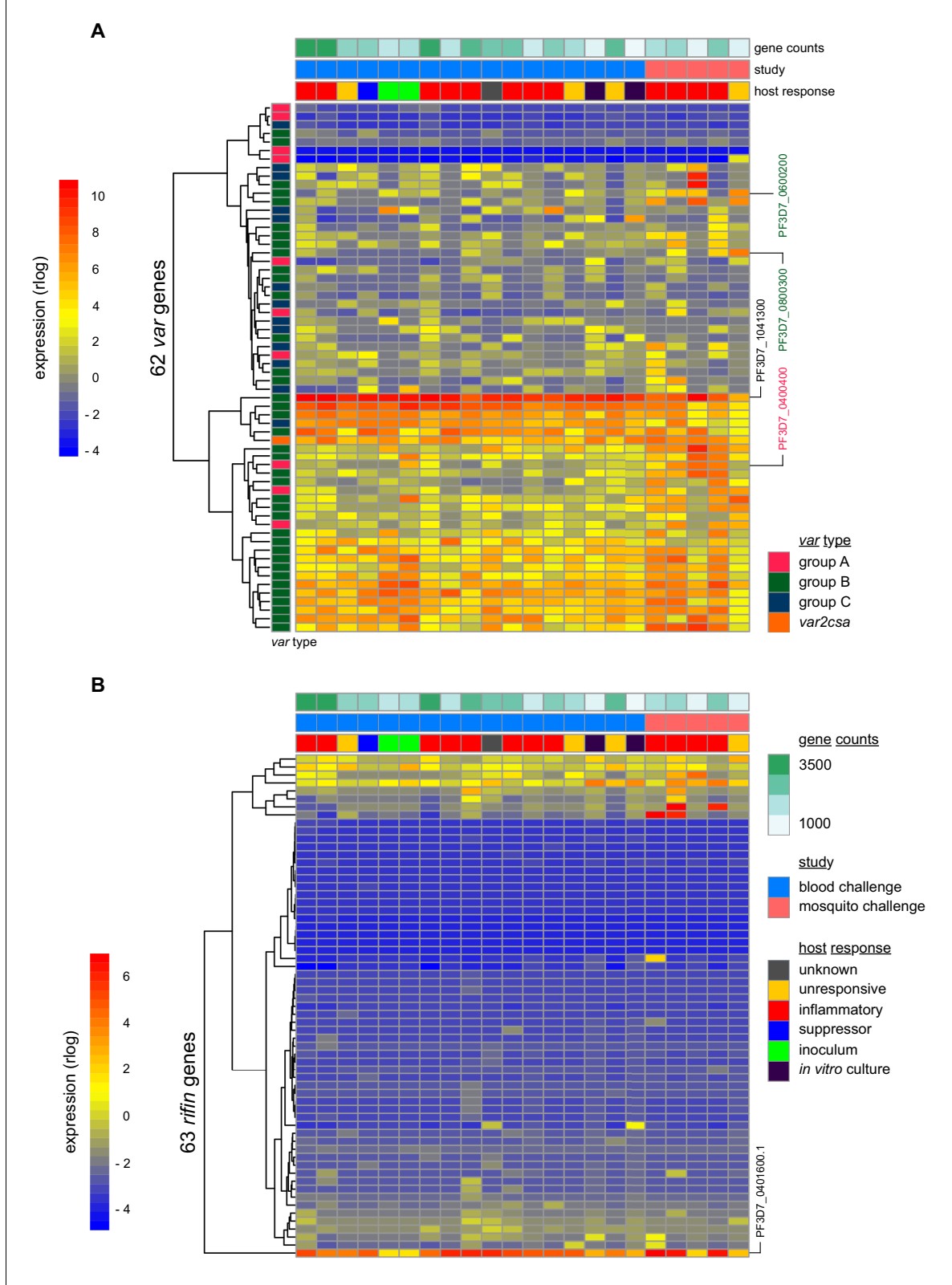

**Figure 4.** Parasite variants associated with severe disease do not rapidly expand in naive hosts. (A and B) Rlog expression values of *var* (A) and *rifin* (B) genes in the inoculum and diagnosis parasite samples after blood challenge (blue study); parasites obtained from volunteers infected by mosquito bite are also shown (pink study). Genes (rows) are ordered by hierarchical clustering and colour-coded by *var* type; parasite samples are colour-coded by host response; and in vitro cultured ring-stage parasites are shown for comparison. Two volunteers did not have parasite sequencing data

*Figure 4 continued on next page*

*Figure 4 continued*

(unresponsive volunteer v208 and suppressor volunteer v022) and the two inoculum samples are technical replicates of one biological sample. The *var* (PF3D7_1041300) and *rifin* (PF3D7_0401600) genes dominantly expressed across all samples are labelled. *Var* genes associated with severe disease (group A variant PF3D7_0400400 and DC8-like variants PF3D7_0600200 and PF3D7_0800300) are also labelled. Gene counts show the number of parasite genes that have at least three uniquely mapping reads; this provides a measure of genome coverage in every sample.

The online version of this article includes the following source data and figure supplement(s) for figure 4:

**Source data 1.** Rlog expression values of *var* and *rifin* genes in parasites isolated from whole blood after controlled human malaria infection.

**Figure supplement 1.** Group B *var* genes dominate the PfEMP1 landscape in naive hosts.

When we expanded our analysis to include *rifin* genes we still could find no evidence of gene switching or selection. Indeed, the only *rifin* that was differentially expressed at diagnosis (PF3D7_0401600, log2 fold-change = 7.59, adj p=0.0002) was already dominantly transcribed at the start of infection (*Figure 4b*). It should be noted, however, that ex vivo RNA-sequencing (ideal for *var* gene analysis as whole blood contains mainly ring-stage parasites) is less suited to *rifins*, which are predominantly transcribed at later stages of the parasite life cycle (*Kyes et al., 2000*). We therefore moved on to directly test the relationship between inflammation and parasite VSA expression. Once again, we used DESeq2 to identify differentially expressed parasite genes but this time between inflammatory volunteers and the rest of the cohort. There were no differentially expressed *var* or *rifin* genes (adj p<0.01). In fact, only one differentially expressed gene could be identified (*Pfck1*) - this gene encodes casein kinase one and was more highly transcribed in the inflammatory group (PF3D7_1136500, log2 fold-change = 7.32, adj p=0.002). The diverse clinical outcomes observed in this trial cannot therefore be easily explained by variation in parasite gene expression; instead, they must largely originate from host-intrinsic immune variation.

## Discussion

In this study, we set out to map immune variation in falciparum malaria and explore the interplay between parasite and host factors in shaping outcome of infection. Of particular interest was how group A and DC8 *var* genes, which are associated with severe malaria in children and adults, come to dominate infections when parasites egress from the liver expressing mainly group B and C *var* genes. One leading hypothesis is that group A/DC8 variants have an intrinsic growth and/or survival advantage leading to their rapid expansion in naive hosts. To directly test this hypothesis, we developed methodology that allowed us to measure *var* gene expression across 10 days of infection in a human blood challenge model. Surprisingly, we found no transcriptional evidence for the preferential expansion of group A or DC8 variants. These data indicate that other factors (such as inflammation, tissue damage, or endothelium activation) must be required to promote the selection of these variants in hosts that develop severe disease. We also found that the diverse outcomes of controlled human malaria infection must largely originate from host-intrinsic variation as just one differentially expressed parasite gene could be identified between volunteers that trigger systemic inflammation (leading to hallmark symptoms of clinical malaria) and those that remain asymptomatic (by preventing or delaying the acute phase response). Taken together, these data emphasise the importance of the human host in determining their own fate.

This conclusion is well supported by epidemiological studies showing that the clinical manifestations of severe malaria depend upon host age (*Dondorp et al., 2008*) and children learn to tolerate high parasite densities, which could otherwise cause life-threatening disease (*Gonçalves et al., 2014*; *Ademolue et al., 2017*). As such, we must understand the rules of immune decision-making if we want to uncouple mechanisms of pathology versus tolerance. In this study, we identified at least three possible early outcomes of blood-stage infection. The most likely outcome is interferon-stimulated inflammation, which is characterised by activation of myeloid cells and systemic release of inflammatory molecules. Notably, only inflammatory volunteers developed hallmark signs and symptoms of malaria (e.g. pyrexia and lymphopenia) providing further support for the idea that inflammation causes the earliest manifestations of disease. It is often assumed that the flip-side to this cost is

immediate or early control of parasite burden and yet we find that the parasite multiplication rate of the inflammatory group is comparable to all other volunteers. It therefore appears that we might need to re-assess the role of innate immunity in the first hours and days of the pathogenic blood cycle – rather than serving as an early brake on parasite replication it may be that systemic inflammation primarily functions to switch bone marrow production in favour of myelopoiesis and mobilise the required effector cells to the spleen.

Recent evidence suggests that blood-stage parasites first trigger interferon signalling in the bone marrow (*Spaulding et al., 2016*) and the transcriptional changes we observe in whole blood are thus likely to be a signature of activated monocytes and neutrophils trafficking from bone marrow to spleen. Furthermore, the major secreted products of this emergency myeloid response (CXCL9 and CXCL10) specifically recruit T cells out of the circulation (*Groom and Luster, 2011*). The innate response to malaria therefore quickly re-structures and co-localises key strands of the immune system; whether this leads to an effective response that can limit parasite burden and minimise collateral damage will depend upon the outcome of cell-cell circuits in the spleen. Measuring the activation and differentiation of CD4$^+$ T cells – the key orchestrators of innate and adaptive immunity – would provide a downstream readout of these critical tissue-specific interactions in human volunteers. Unfortunately, we cannot begin to examine these outcomes in our current datasets because at our final time-point (diagnosis) CD4$^+$ T cells are still sequestered in the tissue. Nevertheless, future studies could extend sampling beyond drug treatment to analyse resolution of the acute phase response and examine the heterogeneity between volunteers in T cell profiles as they are released back into the circulation. Due to the inherent adaptive plasticity of human T cells (*Sallusto et al., 2018*), this is likely to reveal much greater variation between hosts in their response to a first malaria episode.

A second common outcome of infection in this study was immune quiescence. Although the course of infection tended to be shorter in unresponsive volunteers than the others this could not fully explain their lack of response to blood challenge. After all, half of unresponsive volunteers (2/4) had parasite densities comparable to half of the inflammatory group (4/8). And furthermore, v017 triggered a measurable inflammatory response at just 157 parasites ml$^{-1}$, whereas v018 remained unresponsive at 20,000 parasites ml$^{-1}$. These observations are entirely in-line with meta-analysis of historical malariotherapy data that measured the pyrogenic threshold in falciparum malaria as spanning more than four-orders of magnitude (*Molineaux et al., 2002*). Immune variation therefore dictates how you respond and when. Pushing up the pyrogenic threshold to disarm emergency myelopoiesis is a well-recognised host adaptation that promotes immune quiescence and allows individuals to transition from clinical to asymptomatic malaria (usually in adolescence, *Gatton and Cheng, 2002*; *Portugal et al., 2014*). This is helpful if you have developed partial humoral immunity that can restrict total pathogen load but whether a pre-existing high threshold is advantageous in a naive host is unclear. This could indicate a natural tendency towards disease tolerance (*Medzhitov et al., 2012*) or it may simply delay the time until an emergency myeloid response inevitably has to be triggered. Indeed, it remains a possibility that all naive hosts (including those that initially suppress genes involved in myeloid cell activation) will eventually converge on interferon-stimulated inflammation as the infection progresses. After all, every volunteer increased their circulating levels of adenosine as part of a shared metabolic response to blood-stage infection and adenosine is a potent paracrine inhibitor of inflammation (*Faas et al., 2017*). This metabolic switch may therefore represent a hardwired mechanism of self-regulation because systemic inflammation is an unavoidable fate in naive hosts.

We also examined the in vivo relationship between inflammation and parasite VSA expression, asking whether this could shape the clinical outcome of controlled human malaria infection. We confirmed the previous finding that group B *var* genes, which encode PfEMP1 variants associated with uncomplicated malaria, are dominantly expressed in human volunteers infected by mosquito bite (*Bachmann et al., 2019*; *Bachmann et al., 2016*; *Peters et al., 2002*; *Wang et al., 2009*). The unique features of the blood challenge model then allowed us to examine the widely held belief that variants associated with severe disease rapidly and preferentially expand in naive hosts. Unexpectedly, we found no evidence to support this hypothesis. In fact, we found limited transcriptional evidence of *var* gene switching or selection over six cycles of replication. This is in contrast to a previous study that suggested *var* gene expression changes within the first few days of blood-stage infection (*Lavstsen et al., 2005*). However, in that study parasites were isolated from volunteers and

then placed into culture for one month before *var* gene profiling – switching may therefore have occurred in vitro confounding interpretation of these data. Our study, which measured *var* gene expression ex vivo, avoids any bias introduced by culturing parasites but nevertheless has limitations that require careful consideration. In particular, we have studied a single parasite genotype in a small number of hosts over a short window of infection. The potential for these factors to explain the absence of *var* gene switching/selection are considered below.

This study (and most previous CHMI trials) used the 3D7 *P. falciparum* clone or its parent line NF54 (*Stanisic et al., 2018*). The 3D7 clone was cultured for many years before preparation of the inoculum used for blood challenge (*Cheng et al., 1997*) and could have acquired mutations that affect its behaviour in vivo (*Claessens et al., 2017*). This could theoretically include a reduced capacity to switch *var* gene expression; direct experimental evidence that the 3D7 parasites used in this study have retained their ability to switch would therefore strengthen our conclusions. Nevertheless, in the absence of these data it is notable that a previous study using the same 3D7 inoculum clearly showed *var* gene switching when parasites recovered from volunteers were returned to culture (*Peters et al., 2002*). Furthermore, in our mosquito challenge cohort there was considerable variation between volunteers in which group A and DC8 *var* genes were highly transcribed, indicating that 3D7 has retained its potential to express a wide range of virulence-associated variants. Overall, we consider it possible, but unlikely, that our data can be explained by a mutation in 3D7 that affects *var* gene switching. A more interesting possibility is that parasite genotypes may vary in their capacity to switch towards group A or DC8 variants. Additional parasite genotypes are being developed for CHMI that would allow this hypothesis to be tested in vivo (*Moser et al., 2020*; *Stanisic et al., 2015*).

Another possibility is that 3D7 might only switch towards group A/DC8 variants in a minority of hosts. Indeed, in endemic settings the frequency of severe disease is low (around 1% of all clinical episodes) and the small number of volunteers in this study would be unlikely to capture a rare event. Nevertheless, it is clear that the incidence of severe malaria is much higher in naive hosts (*Marsh and Snow, 1997*). And moreover, naive hosts in endemic areas are generally infants and young children who are much less susceptible to severe disease than naive adults (*Baird et al., 2003*; *Baird et al., 1998*). In CHMI, we are therefore recruiting those individuals who are most likely to develop life-threatening complications in the absence of drug treatment; we can therefore reasonably expect that severe disease would not have been a rare outcome of infection in our two cohorts of volunteers. As such, it is unlikely that a different pattern of *var* gene switching would have been observed if a larger number of volunteers were examined, but this cannot be excluded. An alternative explanation is that a parasite density threshold needs to be crossed to promote *var* gene switching in vivo, and this threshold is not reached in controlled human malaria infection. In support of this argument, parasite density is known to be an important factor in severe malaria (*Dondorp et al., 2005*) and this effect could potentially be mediated by quorum sensing, which operates in trypanasomes to control their virulence (*Rojas et al., 2019*). However, no external signals that can cause *var* gene switching have yet been identified (*Deitsch and Dzikowski, 2017*; *Guizetti and Scherf, 2013*).

How then do group A and DC8 *var* genes become predominantly transcribed in patients with severe malaria? The most plausible explanation for our data is that the short duration of CHMI does not allow enough time for the selection of these variants, and that this would have been observed if drug treatment was withheld. However, the preferential expansion of group A/DC8 variants would need to occur rapidly because at the mean parasite multiplication rate of 10 seen in this study only 2–3 more cycles of replication would have been required for some volunteers to reach parasite burdens at which severe disease can occur (5000 parasites $\mu l^{-1}$ or 0.1% parasitaemia [*Field, 1949*]). *Var* gene switching rates of culture-adapted parasites are low in vitro (estimated mean on-rates of 0.025–3% per generation [*Horrocks et al., 2004*; *Roberts et al., 1992*; *Ye et al., 2015*]) and although mathematical modelling suggests the possibility of higher switching rates in vivo (*Gatton et al., 2003*) experimental data are lacking. We therefore propose that rather than a sudden increase in the rate of switching an environmental change instead leads to the rapid selection of variants associated with severe disease – this could include systemic inflammation, collateral tissue damage, metabolic disturbance and/or endothelium activation. In this scenario, group A and DC8 variants have no intrinsic growth or survival advantage; instead, their preferential expansion in naive

hosts relies upon infection-induced changes in their within-host environment. The responsibility for severe malaria is therefore likely shared between parasite and host.

In theory, inflammation will lead to the selection of parasites expressing group A and DC8 *var* genes because of cytokine-mediated upregulation of endothelial receptors such as ICAM-1, which is upregulated in response to TNF (*Bernabeu et al., 2019*). This is predicted to enhance their cytoadherence and reduce mechanical clearance in the spleen, as compared to parasites expressing group B and C *var* genes that bind only CD36 (*Smith et al., 2013*). In our study, there was no evidence for increased transcription of group A or DC8 *var* genes in volunteers who showed clear clinical, transcriptional, and biochemical evidence of inflammation. However, circulating levels of plasma TNF had not yet increased at diagnosis and so more time is clearly required for interferon signalling to drive the cytokine cascades necessary to activate endothelial cells. One argument against an essential role for inflammation is that Endothelial Protein C Receptor (EPCR), currently thought to be a key receptor for cytoadherence in cerebral malaria (*Turner et al., 2013*), is downregulated on microvascular endothelial cells exposed to inflammatory cytokines (*Bernabeu et al., 2019*). The role of EPCR as a receptor for infected red cell binding remains controversial (*Azasi et al., 2018*) but if EPCR does prove to be a major host receptor for sequestration then selection of dual-binders that also interact with ICAM-1 (*Lennartz et al., 2017*) would be expected in an inflammatory environment. One other intriguing finding that emerged from our analysis was that parasites from inflammatory volunteers express higher levels of *Pfck1* – a serine-threonine kinase (*Cheong and Virshup, 2011*; *Jiang et al., 2018*) that has been shown to regulate pathogen virulence in leishmaniasis (*Dan-Goor et al., 2013*) and toxoplasmosis (*Wang et al., 2016*), and which may be able to directly regulate type I interferon signalling (*Liu et al., 2009*). If this observation could be validated in a separate study it would identify a possible mechanism through which malaria parasites may in turn attempt to influence the host response to infection. CHMI trials that follow volunteers for one or two more blood cycles would be required to explore the consequences of increased *Pfck1* transcription and of prolonged systemic inflammation, and consideration should be given as to whether this can be done safely.

To complement the study of *var* gene switching and selection in naive hosts, future work should also aim to examine the neutralisation of parasite variants associated with severe disease in semi-immune adults. Immunity to severe malaria can be acquired early in life (*Gonçalves et al., 2014*) and so will require the rapid production of cross-reactive antibodies that recognise and eliminate group A and DC8 variants if these are sufficient to cause life-threatening disease. Whether antibodies can effectively neutralise these variants in vivo remains to be directly shown, but the RNA-sequencing methodology that we developed for this study (https://doi.org/10.17504/protocols.io.brgjm3un) now allows transcriptional profiling of *P. falciparum* at ultra-low densities – indeed, we can spike just 2000 ring-stage parasites into 50 ml whole blood and map their complete VSA profiles (*Figure 4*). Applying this methodology to CHMI in an endemic setting would therefore allow switching and selection to be tracked under immune pressure in volunteers that can restrict and slow parasite replication.

In summary, our study failed to provide compelling transcriptional evidence that parasite variants associated with severe disease rapidly and preferentially expand in vivo. Our data do not therefore support the hypothesis that group A and DC8 variants have an intrinsic growth and/or survival advantage in naive hosts. This leaves open the question as to how these variants come to dominate severe episodes of malaria when parasites egress from the liver expressing mainly group B and C *var* genes. We propose that infection-induced changes in the host environment (including inflammation) can rapidly select for parasites with a higher efficiency to cytoadhere, sequester and cause severe disease. Notably, our study uncovered marked variation between volunteers in their innate response to blood-stage parasites, which might explain why not all clinical episodes lead to severe malaria. An enormous body of evidence now supports the critical role of host-intrinsic variation in shaping the human immune response to pathogens and their products (*Bakker et al., 2018*; *Brodin et al., 2015*; *Li et al., 2016*; *Patin et al., 2018*; *Piasecka et al., 2018*; *Ter Horst et al., 2016*); remarkably most of this variation is underpinned by non-heritable factors (*Bakker et al., 2018*; *Brodin et al., 2015*; *Piasecka et al., 2018*; *Ter Horst et al., 2016*). Taken together, these data emphasise the importance of the human host in shaping their own fate.

# Materials and methods

**Key resources table**

| Reagent type (species) or resource | Designation | Source or reference | Identifiers | Additional information |
|---|---|---|---|---|
| *Strain, strain background (Plasmodium falciparum)* | clone 3D7 | *Cheng et al., 1997* | | PMID:9347970 |
| Sequence-based reagent | TaqMan probe | Applied Biosystems | | 5' FAM AACAATTGGAGGGCAAG NFQ-MGB 3' |
| Sequence-based reagent | ISPCR primer | biomers.net | | 5'- AAG CAGTGGTATCAACGCAG AGT −3' |
| Sequence-based reagent | LNA-modified TSO | exiqon.com | | 5'- AAGCAGTGGTATCAACG CAGAGTACATrGrG+G −3' |
| Sequence-based reagent | Oligo-dT$_{30}$VN | biomers.net | | 5'- AAGC AGTGGTATCAACGCAGAGT ACT$_{30}$VN −3' |
| Commercial assay or kit | Tempus spin RNA isolation | Applied Biosystems | cat. no. 4378926 | |
| Commercial assay or kit | RNA clean and concentrator | Zymo Research | cat. no. R1016 | |
| Commercial assay or kit | Globin-Zero Gold | Illumina | cat. no. GZG1224 | |
| Commercial assay or kit | RNA 6000 pico chip | Agilent | cat. no. 5067–1513 | |
| Commercial assay or kit | High sensitivity DNA kit bioanalyzer | Agilent | cat. no. 5067–4626 | |
| Commercial assay or kit | TruSeq stranded mRNA library | Illumina | cat. no. RS-122–2101 | |
| Commercial assay or kit | NEBNext Ultra DNA library prep kit | New England Biolabs | cat. no. E7370 | |
| Software, algorithm | Kallisto v0.42.3 | *Bray et al., 2016* | RRID:SCR_016582 | PMID:27043002 |
| Software, algorithm | EdgeR | *Robinson et al., 2010* | RRID:SCR_012802 | PMID:19910308 |
| Software, algorithm | ClueGO v2.5.4 | *Bindea et al., 2009* | RRID:SCR_005748 | PMID:19237447 |
| Software, algorithm | IDEOM | *Creek et al., 2012* | | PMID:22308147 |
| Software, algorithm | mzCloud | | RRID:SCR_014669 | |
| Software, algorithm | DESeq2 | *Love et al., 2014* | RRID:SCR_015687 | PMID:25516281 |
| Other | Saponin from quillaja bark | Sigma Aldrich | cat. no. S7900 | test each batch |
| Other | Leucoflex LXT filters | Macopharma | | macopharma.com /transfusion |
| Other | KAPA HiFi HotStart ReadyMix | Biosystems | cat. no. KK2601 | |
| Other | AMPure XP beads | Beckman Coulter | cat. no. A63881 | |

## Study participants and ethical approval

The 14 volunteers recruited for blood challenge were infectivity controls in a phase I/IIa vaccine trial (VAC054) that took place at the Centre for Clinical Vaccinology and Tropical Medicine (CCVTM),

Oxford (*Payne et al., 2016*). This study received ethical approval from the UK NHS Research Ethics Service (Oxfordshire Research Ethics Committee A, 13/SC/0596) and the Western Institutional Review Board (WIRB) in the USA (20131985). The study was approved by the UK Medicines and Healthcare products Regulatory Agency (MHRA) (21584/0326/001–0001) and the trial was registered on ClinicalTrials.gov (NCT02044198). The five volunteers infected by mosquito bite were recruited at the same site and this study (VAC065) received ethical approval from the UK NHS Research Ethics Service (South Central Berkshire Research Ethics Committee, 16/SC/0261) and was approved by the UK MHRA (21584/0360/001–0001). The trial was registered on ClinicalTrials.gov (NCT02905019). Both trials were conducted according to the principles of the current revision of the Declaration of Helsinki 2008 and in full conformity with the ICH guidelines for Good Clinical Practice (GCP).

## Controlled human malaria infection

### Blood challenge
Details of inoculum preparation and assessment of parasite viability have been published previously (*Payne et al., 2016*). The 14 volunteers (non-vaccinated infectivity controls) received an identical challenge by intravenous injection of 690 *P. falciparum* (clone 3D7) infected erythrocytes in 5 ml of 0.9% saline. All inoculations were performed within 2 hr and 13 min of inoculum preparation.

### Mosquito challenge
CHMI was performed as previously described (*Sheehy et al., 2012*) using five infectious bites from *P. falciparum* (clone 3D7) infected *Anopheles stephensi* mosquitoes at the Alexander Fleming Building, Imperial College, London, UK. Infected mosquitoes were supplied by Jittawadee R. Murphy, Department of Entomology, Walter Reed Army Institute of Research, Washington, DC, USA.

## Monitoring volunteers

Blood samples were collected to measure parasitaemia by qPCR twice daily, starting 2 days post-infection for blood challenge and 6.5 days post-infection for mosquito challenge. In both trials, thick blood smears were also evaluated by experienced microscopists at each time-point. All reported clinical symptoms (pyrexia, fever, rigor, chills, sweats, headache, myalgia, arthralgia, back pain, fatigue, nausea, vomiting, and diarrhoea) were recorded as adverse events and assigned a severity score: 1 - transient or mild discomfort (no medical intervention required); 2 - mild to moderate limitation in activity (no or minimal medical intervention required); 3 - marked or severe limitation in activity requiring assistance (may require medical intervention). A cumulative clinical score was then obtained by summing these adverse events across all time-points for each volunteer. At every clinic visit, staff at the CCVTM also assessed core temperature, heart rate and blood pressure. And on days −1, 6, 28, and 90 post-infection (plus diagnosis) full blood counts and blood biochemistry were performed at the Churchill and John Radcliffe Hospitals in Oxford, providing five-part differential white cell counts and a quantitative assessment of electrolytes, urea, creatinine, bilirubin, alanine aminotransferase, alkaline phosphatase and albumin.

## Diagnostic criteria and drug treatment

Treatment with the artemether and lumefantrine combination drug Riamet was prescribed to volunteers when two out of three diagnostic criteria were met: a positive thick blood smear and/or parasite density >500 parasites ml$^{-1}$ by qPCR and/or symptoms consistent with malaria.

## qPCR and parasite multiplication rate modelling

Blood was collected and prepared for qPCR analysis as previously described (*Payne et al., 2016*; *Sheehy et al., 2012*). In brief, DNA was extracted from 0.5 ml whole blood using the Qiagen Blood Mini Kit (as per manufacturer's instructions) and 10% of each extraction was run in triplicate wells (equating to 150 µl total blood volume). Previously published primers with TaqMan probes (5′ FAM-AAC AAT TGG AGG GCA AGNFQ-MGB 3′) (Applied Biosystems) were used to amplify the *P. falciparum* 18S ribosomal region and parasites ml$^{-1}$ equivalent mean values were calculated using a defined plasmid standard curve and TaqMan absolute quantitation. Mean values below 20 parasites ml$^{-1}$ (or values with only one positive replicate of the three tested) were classed as negative. Parasite multiplication rate (PMR) was then calculated using a linear model fitted to log10 transformed

qPCR data, as previously described (*Sheehy et al., 2012*; *Douglas et al., 2013*). In the blood challenge model, fitted lines were constrained to pass through the known starting parasitemia, determined by the viability of the inoculum and a weight-based estimate of each volunteer's total blood volume (*Payne et al., 2016*).

## Uninfected control volunteers
### Blood challenge
Four healthy adult volunteers were recruited at the University of Edinburgh to serve as uninfected controls for the blood challenge study. These volunteers had no history of malaria, and included two male and two female Caucasians with a mean age of 24 years (range 22–25). Written informed consent was given by all participants and the study was approved by the University of Edinburgh, School of Biological Sciences Ethical Review Committee (arowe-0002 and pspence-0002).

### Mosquito challenge
One volunteer recruited to the VAC065 study was exposed to the bites of 5 infectious mosquitoes but did not develop measurable parasitaemia at any point during the 28 day study (limit of detection is five parasites ml$^{-1}$). This volunteer therefore served as an internal uninfected control for the mosquito challenge study.

## Processing whole blood for host RNA and plasma
### Blood challenge
For microarray analysis, 3 ml venous blood was drawn directly into a Tempus Blood RNA Tube (Applied Biosystems), mixed and stored at −80˚C. To obtain plasma, 14 ml venous blood was drawn into lithium heparin vacutainers, transferred to Leucosep tubes (Greiner Bio-One) containing 15 ml Lymphoprep (Axis Shield) and centrifuged at 1000xg for 13 min at room temperature. Two ml of the plasma fraction was collected, snap-frozen on dry ice and stored at −80˚C.

### Mosquito challenge
For host RNA-sequencing, 6 ml venous blood was drawn into EDTA-coated vacutainers, from which 1 ml blood was removed and mixed with 2 ml Tempus reagent; samples were stored at −80˚C. To obtain platelet-depleted plasma, a further 1 ml blood was removed from the vacutainer and given two spins; first at 1000xg for 10 min and then at 2000xg for 15 min (both at 4˚C). After each spin, the plasma was transferred to a new tube (leaving behind any residual red cells or pelleted platelets) and finally the plasma was snap-frozen on dry ice and stored at −80˚C.

## Host RNA extraction, quantification, and quality control
RNA extraction was performed with the Tempus Spin RNA Isolation Kit (Applied Biosystems) according to the manufacturer's instructions. The following modification was applied to mosquito challenge samples because of their reduced volume: after thawing, just 1 ml PBS was added to each sample to maintain Tempus stabilising reagent at the correct final concentration. Note that for all samples, a DNase treatment step using Absolute RNA wash solution (Applied Biosystems) was included for the removal of genomic DNA. Samples were quantified by nanodrop and RNA integrity was assessed on an Agilent 2100 Bioanalyzer using RNA 6000 nano chips. A RIN value above 7.0 was accepted as sufficiently high-quality RNA for downstream steps.

## Host microarray and data processing (blood challenge)
Whole blood transcriptomics was undertaken with the Affymetrix GeneChip Human Transcriptome Array 2.0 ST by Hologic Ltd. (Manchester, UK); samples were randomised to ensure unbiased sample-chip positioning. Raw array data in the form of compressed CEL files were processed in the R environment using Bioconductor packages (oligo, pd.hta.2.0, affy) to generate an HTAFeatureSet object (gene level). After visual inspection for QC purposes, raw and normalised (using the rma() function of the affy package or normalizeQuantiles() function from the limma Bioconductor package, as appropriate) versions of the intensities were retained for subsequent analyses. Array feature annotation was generated using the AnnotationForge package and the Bioconductor human reference database (org.Hs.eg.db).

### Plotting the 100 protein-coding genes with highest variance

For each volunteer, a subset of the data was made encompassing all available time-points in chronological order. For each array feature, the variance of the intensities across the time-points was calculated using the var() function in R. The 100 protein-coding array features with the highest variance were then selected and for each of these features the median value of the intensities across all time-points was calculated and subtracted from the actual intensity values, resulting in a set of 'deviation from median' values. These values were plotted using the heatmap.2() function in R and dendrograms, where shown, were calculated based on Euclidean distance measures of the input data (hclust() and dist() functions of the stats package).

### Plotting the 517-gene superset

The lists of 100 protein-coding genes with highest variance in each of the 14 volunteers infected by blood challenge were merged to create a non-redundant set of 517 unique genes. Log2 transformed intensity values were then scaled with scale (scale=TRUE, center=TRUE), plotted with pheatmap and clustered using hclust and cutree in R, with k = 8.

### PCA plots of the 517-gene superset

For each volunteer, a subset of the data was made encompassing all available time-points in chronological order. The log2 transformed intensity values of the 517 genes were normalised using the normalizeQuantiles()function from the limma Bioconductor package prior to principal component analysis using the prcomp() function of the stats Bioconductor package. Plots of the first two dimensions were generated using the standard plot() function of the graphics package. To measure the magnitude of each volunteer's immune response we calculated the Euclidean distance travelled along principal component 1, as this accounted for almost all of the variance in every time-course. As every sample in a time-course was centred around zero we calculated the distance travelled for every volunteer (relative to their own average position through time) as the x-coordinate of the diagnosis sample.

### Differential gene expression

Analysis of differential gene expression was carried out using the limma package from Bioconductor, with eBayes correction for multiple testing; an adjusted p value below 0.05 was required for significance. This analysis was performed to quantify the number of differentially expressed genes between time-points for each group (unresponsive, inflammatory, and suppressor). It was also carried out to quantify the number of differentially expressed genes between groups at a given time-point.

## Allocation of volunteers to groups after blood challenge

To be classified as responsive to blood challenge volunteers had to pass two sequential thresholds: first, for every volunteer the mean variance of their 50 most variable protein-coding genes was calculated. This figure had to be at least 1.5-fold higher than the mean of the four uninfected controls (i.e. $1.5 \times 0.202 = $ **0.303**). Second, the distance travelled in each volunteer's PCA plot of the 517-gene superset had to be greater than two standard deviations above the mean of the four uninfected controls (i.e. $2.67 + (2 \times 1.102) = $ **4.874**). Volunteers had to pass both thresholds in order to be classified as responsive; those who failed one or both of these tests (v018, v012, v208, and v020) were classified as unresponsive. The remaining volunteers were then grouped into inflammatory (v016, v017, v013, v027, v026, v206, v106, and v024) or suppressor (v022 and v019) hosts based on their upregulation of clusters 2 and 4 or downregulation of clusters 1 and 3, respectively (see *Figure 1*). Note that differential gene expression analysis (as detailed above) was used to confirm that the unresponsive group had zero differentially expressed genes at diagnosis (compared to baseline).

## Host RNA-sequencing and data processing (mosquito challenge)

Indexed sequencing libraries were prepared from 500 ng RNA using the TruSeq Stranded mRNA Library kit (Illumina), according to the manufacturer's instructions. Samples were pooled and sequenced on two separate lanes of the same Illumina HiSeq v4 flow cell to generate ~33 million reads per sample (75 bp paired-end reads). Note that globin depletion was not performed; instead,

sequencing was carried out to sufficient depth that the globin reads (accounting for no more than 20% of any given sample) could simply be discarded during data processing. Read counts per gene were quantified against the Gencode human v28 basic transcript sequences (*Frankish et al., 2019*) using Kallisto v0.42.3 (92). The index was built using default parameters and quantification was performed with -t 12 -b 0; genes not expressed with at least one count per million in at least one sample were excluded. EdgeR (*Robinson et al., 2010*) was used to identify highly dispersed genes (BCV >0.4 and FDR < 0.1) for each volunteer's time-course with the exact test. Next, the lists of highly dispersed genes in each of the five volunteers infected by mosquito bite were merged to create a non-redundant set of 226 unique genes, which were used for clustering with hclust in the pheatmap R package. This identified a single major cluster comprising 117 unique genes that was taken forward for downstream analyses (e.g. plotting the 117-gene superset and ClueGO). Note that to visualise and cluster the 226 unique genes and the 117-gene superset, read counts across the entire dataset were normalised using rlogs (blind=TRUE) in DESeq2 (*Love et al., 2014*).

## Gene ontology analysis and networks (ClueGO)
### Blood challenge
The list of 2028 genes differentially expressed at diagnosis in the inflammatory group was imported into ClueGO v2.5.4 (*Bindea et al., 2009*; *Mlecnik et al., 2014*). ClueGO identified the significantly enriched GO terms associated with these genes and placed them into a functionally organised non-redundant gene ontology network based on the following parameters: pvalue cutoff = 0.05; correction method used = Bonferroni step down; min. GO level = 5; max. GO level = 11; number of genes = 3; min. percentage = 5.0; GO fusion = true; sharing group percentage = 40.0; merge redundant groups with >40.0% overlap; kappa score threshold = 0.4; and evidence codes used [All]. Each of the 34 functional groups was assigned a unique colour and a network was then generated using an edge-weighted spring-embedded layout based on kappa score. To provide an overview of the inflammatory response we then labelled the leading GO term in the top 12 functional groups, as follows: groups were first ordered by adj p value; starting with the most significant group we chose the level 4 GO term with the highest significance and made this the group name; we then moved on to the next most significant group and again chose the level 4 GO term with the highest significance as group name; we repeated this process until 12 groups had been named. Note that there were two exceptions to these rules: first, if there was not a level 4 GO term within a group then this group was excluded (to avoid naming groups that were too broad or specific in ascribing function); and second, if the most significant level 4 GO term was shared with a group already named then this GO term was excluded and the group was instead named after the next most significant level 4 GO term (to avoid redundancy).

### Mosquito challenge
The 117-gene superset was imported into ClueGO to identify the significant GO terms associated with these genes. This was carried out exactly as described for blood challenge with the exception of the following parameter changes: number of genes = 2; min. percentage = 4.0; merge redundant groups with >50.0% overlap.

## Quantification of cytokines and chemokines in plasma
Interferon alpha (IFNα), interferon gamma (IFNγ), CXCL9 (MIG), and CXCL10 (IP-10) were quantified in plasma as part of a custom-design multiplex assay from BioLegend (LegendPlex). Samples were run in duplicate and the assay was performed according to the manufacturer's instructions using low protein-binding filter-bottom plates (Merck). Beads were acquired on a BD LSRFortessa running FACSDiva software and data were analysed with LEGENDPlex analysis software. For blood challenge, data are presented as fold-change relative to pre-infection samples; when analytes were undetectable samples were simply assigned the largest integer below the limit of detection to allow normalisation and graphing of all time-points.

## Quantification of Angiopoietin-2 in plasma
Angiopoietin-2 was measured in plasma at pre-infection and diagnosis time-points using the Human Angiopoietin-2 Quantikine ELISA kit from R and D Systems. Samples were run in duplicate and the

assay performed as per the manufacturer's instructions. Optical density (OD) was determined on a Multiskan Ascent plate reader by subtracting measurements taken at 570 nm wavelength from readings at 450 nm. All samples and standards were background-corrected by subtracting the mean OD measured in blank wells containing only assay diluent. A seven-point standard curve was then constructed by plotting the mean absorbance of each standard against known concentration and using regression analysis to draw a best fit line ($R^2$ = 0.998). Data are presented as mean concentration of the duplicate samples.

## Detection of anti-Cytomegalovirus IgG in plasma

Cytomegalovirus (CMV) seropositivity was assessed in all volunteers before CHMI using an anti-CMV IgG Human ELISA Kit (Abcam). Samples were run in duplicate and OD was determined on a Multiskan Ascent plate reader by subtracting measurements taken at 620 nm wavelength from readings at 450 nm. All samples and controls were background-corrected by subtracting the mean OD measured in substrate blank wells. Samples were scored positive if their absorbance value was > 10% above the mean absorbance value of the CMV IgG cut-off control. This was calculated by converting OD values to Standard Units using the following equation: mean sample absorbance value x 10 / mean absorbance value of the cut-off control. Samples with < 9 standard units were classified as CMV-negative, whereas samples with > 11 standard units were classified as CMV-positive.

## LC-MS-based analysis of plasma metabolites

Malaria-induced changes in metabolite abundance were measured in seven volunteers from the blood challenge cohort: two unresponsive volunteers (v018 and v208), the three most inflammatory volunteers (based on upregulation of plasma CXCL10 and IFNγ) (v016, v017, and v013) and the two suppressor volunteers (v022 and v019). Samples were prepared by diluting 25 µl of plasma in 1000 µl of 4°C (HPLC-grade) chloroform/methanol/water at a 1:2:1 ratio. Samples were then vortexed for 5 min and spun at 13,000xg for 3 min at 4°C. After centrifugation, the supernatant was collected and stored in cryovials at −80°C ready for downstream processing. LC-MS-based metabolomics analyses were conducted at Glasgow Polyomics. Hydrophilic interaction liquid chromatography (HILIC) was performed on a Dionex UltiMate 3000 RSLC system (Thermo Fisher Scientific, UK) using a ZIC-pHILIC column (150 mm × 4.6 mm, 5 µm column from Merck Sequant). For each sample, a 10 µl volume was injected into the system. Samples were maintained at 5°C prior to injection. The column was kept at 30°C throughout the analysis and samples were eluted with a linear gradient (from 20% 20 mM ammonium carbonate in water (A) and 80% acetonitrile (B) to 80% A and 20% B) at a flowrate of 300 µl/min over 24 min. Mass spectrometry analysis was performed on a Thermo Orbitrap QExactive operated in polarity-switching ionisation mode with the following parameters: resolution of 70,000; automated gain control of 106; m/z range of 70–1,050; sheath gas flowrate of 40 arbitrary units (au); auxiliary gas flowrate of 5 au; sweep gas flowrate of 1 au; probe temperature of 150°C and capillary temperature of 320°C. The parameters for positive mode ionisation were as follows: source voltage +3.8 kV, S-Lens RF Level 30.00, S-Lens Voltage −25.00 V, Skimmer Voltage −15.00 V, Inject Flatapole Offset −8.00 V, Bent Flatapole DC −6.00 V. For negative mode ionisation, a source voltage of −3.8 kV was used. Volunteer samples were run alongside three mixes of authentic standards (total number: 248) involved in various metabolic pathways to facilitate metabolite identification based on exact mass and retention time. Fragmentation of the top 20 ions was performed with the following parameters: collision energy: 25%; isolation window: 1.2; dynamic exclusion after one time; exclusion duration: 6s; exclude isotopes: true; and minimum intensity: 5000. LC-MS raw data were processed with IDEOM (using default parameters [*Creek et al., 2012*]), which uses the XCMS (*Smith et al., 2006*) and mzMatch (*Scheltema et al., 2011*) software in the R environment. The levels of reliability of the spectral assignment to metabolites, as defined by the Metabolomics Standard Initiative (*Sumner et al., 2007*), are as follows: 'MSI:1 (identified metabolites)' – high-resolution mass (three ppm) and retention time (5%) matched to an authentic standard and 'MSI:2 (putatively annotated compounds)' – high-resolution mass matched to a public library (three ppm). Xcalibur software (Thermo Fisher Scientific) was used to generate fragmentation spectra, which were further exported into mzCloud to search for compounds in the database with matching spectra.

## Metabolomics data processing and analysis

Metabolite intensities were log2 transformed and intensity ratios were calculated as the fold-change in metabolite abundance at day 8 or diagnosis relative to pre-infection. A pairwise comparison (t-test) between post- and pre-infection samples was then carried out for every metabolite and p values were adjusted for multiple testing by applying a false discovery rate (FDR). Differentially abundant metabolites were called using an FDR threshold of 0.1 and an intensity ratio equivalent to a 1.5-fold change in abundance. A volcano plot showing the intensity ratios and FDR-corrected p values of all metabolites was generated using ggplot2. Next, the differentially abundant metabolites were range-scaled according to *van den Berg et al., 2006* and a heatmap of range-scaled intensity values was generated using the gplots R package. Here range-scaling is used to make every metabolite equally important by scaling the intensity of each metabolite relative to its own biological range across all samples.

## Isolation of parasites for ex vivo RNA-sequencing

A step-by-step guide to our ultra-low density RNA-sequencing methodology for human malaria parasites is available at protocols.io (dx.doi.org/10.17504/protocols.io.brgjm3un).

### Blood challenge

50 ml whole blood was drawn into lithium heparin vacutainers at diagnosis (immediately before drug treatment) and white cells were removed by passing the blood through a leucoflex LXT filter (Macopharma). The flow-through was collected into a 250 ml centrifuge bottle (with silicone O-ring) and the filter was washed with 200 ml PBS to maximise recovery of infected red cells. The flow-through was spun at 1000xg for 10 min at room temperature with brake off and the supernatant carefully aspirated (being careful not to disturb the pelleted red cells). The red cells were gently agitated to resuspend and then lysed by addition of 200 ml ice-cold saponin at 0.015% in PBS (w/v). The cell suspension was incubated on ice for 10 min and then spun at 15,000xg for 10 min at 4°C with brake off to pellet the free parasites. The supernatant was slowly aspirated being very careful not to disturb the pelleted parasites, which were then gently resuspended in 1 ml ice-cold PBS and transferred to a 1.5 ml tube. The parasites were pelleted by centrifugation at 16,000xg for 5 min at 4°C in a microfuge and two sequential washes were performed with 1 ml ice-cold PBS to remove free globin. After the final spin, the parasite pellet was resuspended in 1 ml TRIzol, incubated in a 37°C water bath for 5 min and snap-frozen on dry ice. All samples were stored at −80°C prior to RNA extraction.

### Mosquito challenge

Parasites were isolated from 20 ml whole blood drawn on days 9, 10, and 11 post-infection according to the blood challenge protocol, with the following modifications. Red cells were lysed by addition of 80 ml ice-cold saponin at 0.0075% in PBS (w/v); after the 10 min incubation on ice, free parasites were then pelleted by centrifugation at 18,000xg for 20 min at 4°C with brake off.

### Inoculum

Two samples of the inoculum used for blood challenge were prepared independently for RNA-sequencing to provide technical replicates. For each sample, 1 ml of the pre-diluted inoculum (estimated to contain 1000 parasites ml$^{-1}$ in 0.9% saline) was transferred to a 1.5 ml tube; infected red cells were pelleted by centrifugation at 1000xg for 5 min at room temperature and (after removal of the supernatant) resuspended in 1 ml TRIzol. The samples were carefully mixed, incubated in a 37°C water bath for 5 min and then snap-frozen on dry ice.

### In vitro cultured ring-stage parasites

Two mock samples were prepared using in vitro cultured parasites (clone 3D7) to test the sensitivity of our ultra-low input methodology for parasite RNA-sequencing. Briefly, parasites were cultured and synchronised to ring-stage using 5% sorbitol, and the first mock sample was prepared by spiking 2000 parasites into 50 ml whole blood (40 parasites ml$^{-1}$). The second mock sample was prepared by spiking 8000 parasites into 20 ml whole blood (400 parasites ml$^{-1}$). These parasite densities were chosen to represent the likely burden in the first and second blood cycle after liver egress in

volunteers infected by mosquito bite (*Roestenberg et al., 2012*). Parasites were isolated from these mock samples according to the mosquito challenge protocol above.

## Parasite RNA extraction and depletion of globin and ribosomal RNA

Samples were thawed at room temperature and 200 µl bromochloropropane was added; samples were then vortexed, incubated for 3 min at room temperature and centrifuged at 12,000xg for 15 min at 4°C. 500 µl of the aqueous phase was transferred to a 1.5 ml tube and diluted 1:1 with absolute ethanol. RNA purification was then carried out using the RNA Clean and Concentrator kit, as per the manufacturer's instructions (Zymo Research). RNA was eluted in 14 µl DEPC-treated water and quantified by nanodrop; quality was assessed using the Agilent 2100 Bioanalyzer with RNA 6000 pico chips. A RIN value above 7.0 was accepted as sufficiently high-quality RNA for downstream steps. Samples were then DNase-treated and depleted of globin and ribosomal RNA with the Globin-Zero Gold kit (Illumina), according to the manufacturer's instructions. For samples with less than 1 µg RNA (the recommended starting material) reagent volumes were downscaled for low-input methodology (as described in the kit). After removal of globin and ribosomal RNA, the remaining RNA was precipitated overnight at 4°C in 600 µl absolute ethanol, 18 µl sodium acetate (3M), and 2 µl linear acrylamide (5 mg/ml). The RNA was then pelleted by centrifugation at 12,000xg for 20 min at 4°C and washed two times with 75% ethanol. Samples were air-dried for 5 min and the RNA was dissolved in 12 µl DEPC-treated water by heating to 65°C for 5 min. RNA was stored at −80°C prior to cDNA synthesis and amplification.

## Parasite RNA-sequencing and data processing

### Blood challenge

Double-stranded cDNA was synthesised from 2 µl parasite RNA using the Smart-Seq2 protocol described by *Picelli et al., 2013* and modified for use with low-input *P. falciparum* RNA (*Reid et al., 2018*). cDNA was amplified with 25 cycles of PCR using the KAPA HiFi reaction mix and ISPCR primer, and purified with Agencourt AMPure XP beads at a 1:1 DNA to bead ratio. Indexed sequencing libraries were constructed from 500 ng of cDNA with no further PCR amplification. cDNA was sheared to 400–600 bp with a Covaris sonicator and libraries prepared using the NEBNext Ultra DNA library prep kit, according to the manufacturer's instructions. Libraries were pooled and run on two separate lanes of distinct Illumina HiSeq v4 flowcells to produce 75 bp paired-end reads. Read counts per gene were quantified against the *P. falciparum* v3 transcript sequences (*Böhme et al., 2019*) using Kallisto v0.42.3 (*Bray et al., 2016*). The index was built using default parameters and quantification was performed with -t 12 -b 0.

### Mosquito challenge

Double-stranded cDNA was synthesised and amplified exactly as described for blood challenge, with the exception that the number of PCR cycles was reduced to 20. Indexed sequencing libraries were prepared from 1 ng cDNA with the Nextera XT library kit (Illumina), as per the manufacturer's instructions. One µl of each undiluted library was run on the Agilent bioanalyzer high sensitivity DNA chip for quantification and quality assessment, and all libaries were pooled and run on a single lane of an Illumina Hiseq v4 flowcell. Read counts per gene were quantified against the *P. falciparum* v3 transcript sequences using Kallisto v0.42.3, as above. The index was built using default parameters and quantification was performed with -t 12 -b 0.

## Analysis of parasite *var* and *rifin* expression

For every volunteer infected by mosquito bite, read counts were pooled from days 9, 10, and 11 post-infection to generate a comprehensive parasite transcriptome 2–3 cycles after liver egress. The two inoculum samples (blood challenge) were not pooled and thus provide technical replicates of a single biological sample. Data from the mosquito and blood challenge studies were merged and rlog values (blind=TRUE) were calculated across the entire dataset using DESeq2 (*Love et al., 2014*); these values were then used to generate heatmaps of expression intensity. Sub-family classifications for the 62 *var* genes were determined from PlasmoDB (*Warrenfeltz et al., 2018* based on *Rask et al., 2010*) and 158 *rifin* genes were identified from the same database. Heatmaps were drawn using pheatmap, clustered with hclust. To generate pie charts of *var* gene sub-family

expression we summed FPKM-normalised (Fragments Per Kilobase per Million mapped reads) read counts for each *var* gene sub-family for each volunteer. Genes with a summed FPKM of <1 in any given sample were excluded for that sample and plots were drawn using matplotlib in Python. To examine reads mapping across *var* gene introns, we followed the protocol outlined in *Reid et al., 2018*. Briefly, we generated a HISAT2 (v2.0.0-beta) index of the *P. falciparum* v3 genome sequence using default parameters and mapped reads with parameters -max-intronlen 5000 p 12 (*Kim et al., 2015*). We identified reads overlapping annotated *var* genes using bedtools intersect (*Quinlan and Hall, 2010*). We then selected reads with an *N* in the CIGAR string, indicating a split read. We counted only those reads that were split exactly over the intron of a *var* gene. We called expression for a *var* gene where there were at least two reads mapping over the intron; read counts were then converted to logs of Counts Per Million (CPM) using the total number of reads mapping to all other features and the heatmap was drawn using pheatmap.

### Calling differentially expressed parasite genes

To determine whether any parasite genes were differentially expressed between volunteers with different clinical outcomes after blood challenge, we used DESeq2 (*Love et al., 2014*) to compare parasite gene expression profiles from inflammatory volunteers (n = 8) versus the rest (unresponsive and suppressor, n = 4). In the same way, we used DESeq2 to compare parasite gene expression profiles between the inoculum samples (n = 2) and all volunteer samples (n = 12) after blood challenge.

### Accession numbers

- Blood challenge human microarray dataset (NCBI GEO): GSE132050
- Mosquito challenge human RNA-sequencing dataset (EBI EGA): EGAS00001003766
- Parasite RNA-sequencing dataset (blood and mosquito challenge) (EBI ENA): PRJEB33557
- Blood challenge metabolomics dataset (MetaboLights): MTBLS1188

## Acknowledgements

We are grateful to Julie Furze, Sean Elias, Katie Ewer, the VAC054 study team and the VAC065 study team (Jenner Institute Laboratories and the Centre for Clinical Vaccinology and Tropical Medicine, University of Oxford) for assistance and also for access to samples from these CHMI clinical trials (originally funded by the PATH Malaria Vaccine Initiative, US Agency for International Development (USAID) and the European Union Seventh Framework Programme (FP7/2007-2013) under the grant agreement for MultiMalVax (number 305282)). We also thank all of the clinical trial volunteers and blood donors who participated in this study. This project was supported by the Wellcome Trust-University of Edinburgh Institutional Strategic Support Fund. KM is the recipient of a Medical Research Council PhD studentship (grant no. G40270). AJR is funded by the Medical Research Council (programme grant no. MR/M003906/1) and the Wellcome Sanger Institute is funded by the Wellcome Trust (grant WT206194). AOT is the recipient of a Wellcome Trust PhD studentship (grant no. 203783/Z/16/Z). MPB is part of the Wellcome Centre for Integrative Parasitology (grant no. 104111/Z/14/Z). AVSH and SJD are Jenner Investigators. SJD is the recipient of a Wellcome Trust Senior Fellowship (grant no. 106917/Z/15/Z) and is a Lister Institute Research Prize Fellow. JAR is supported by the Wellcome Trust (grant no. 084226). And PJS is the recipient of a Sir Henry Dale Fellowship jointly funded by the Wellcome Trust and the Royal Society (grant no. 107668/Z/15/Z).

## Additional information

### Funding

| Funder | Grant reference number | Author |
|--------|----------------------|--------|
| Medical Research Council | G40270 | Kathryn Milne |
| Medical Research Council | MR/M003906/1 | Adam J Reid |
| Wellcome Trust | WT206194 | Adam J Reid |

| | | Magda E Lotkowska |
| | | Geetha Sankaranarayanan |
| | | Mandy J Sanders |
| | | Matthew Berriman |
| Wellcome Trust | 203783/Z/16/Z | Aine O'Toole |
| Wellcome Trust | 104111/Z/14/Z | Michael Barrett |
| Wellcome Trust | 106917/Z/15/Z | Simon J Draper |
| Wellcome Trust | WT084226 | Alexandra Rowe |
| Wellcome Trust | 107668/Z/15/Z | Philip J Spence |

The funders had no role in study design, data collection and interpretation, or the decision to submit the work for publication.

## Author contributions

Kathryn Milne, Formal analysis, Validation, Investigation, Visualization, Methodology, Writing - review and editing; Alasdair Ivens, Adam J Reid, Data curation, Software, Formal analysis, Validation, Visualization, Writing - review and editing; Magda E Lotkowska, Geetha Sankaranarayanan, Diana Munoz Sandoval, Investigation, Methodology; Aine O'Toole, Wiebke Nahrendorf, Software, Formal analysis, Validation, Visualization; Clement Regnault, Data curation, Software, Formal analysis, Validation, Visualization; Nick J Edwards, Formal analysis, Validation, Investigation; Sarah E Silk, Project administration; Ruth O Payne, Angela M Minassian, Navin Venkatraman, Resources, Investigation, Project administration; Mandy J Sanders, Resources, Project administration; Adrian VS Hill, Resources, Supervision, Funding acquisition, Project administration; Michael Barrett, Matthew Berriman, Resources, Supervision, Project administration; Simon J Draper, Resources, Supervision, Funding acquisition, Project administration, Writing - review and editing; J Alexandra Rowe, Conceptualization, Formal analysis, Supervision, Funding acquisition, Visualization, Writing - original draft, Writing - review and editing; Philip J Spence, Conceptualization, Formal analysis, Supervision, Funding acquisition, Validation, Investigation, Visualization, Methodology, Writing - original draft, Writing - review and editing

## Author ORCIDs

Nick J Edwards (iD) http://orcid.org/0000-0002-7030-7839
Matthew Berriman (iD) http://orcid.org/0000-0002-9581-0377
Simon J Draper (iD) http://orcid.org/0000-0002-9415-1357
J Alexandra Rowe (iD) https://orcid.org/0000-0002-7702-1892
Philip J Spence (iD) https://orcid.org/0000-0002-5506-2773

## Ethics

Clinical trial registration ClinicalTrials.gov accession codes NCT02044198 and NCT02905019.
Human subjects: VAC054 received ethical approval from the UK NHS Research Ethics Service (Oxfordshire Research Ethics Committee A, ref 13/SC/0596) and the Western Institutional Review Board (WIRB) in the USA (ref 20131985). The study was approved by the UK Medicines and Healthcare products Regulatory Agency (MHRA) (ref 21584/0326/001-0001) and the trial was registered on ClinicalTrials.gov (NCT02044198). VAC065 received ethical approval from the UK NHS Research Ethics Service (South Central Berkshire Research Ethics Committee, ref 16/SC/0261) and was approved by the UK MHRA (ref 21584/0360/001-0001). The trial was registered on ClinicalTrials.gov (NCT02905019). Both trials were conducted according to the principles of the current revision of the Declaration of Helsinki 2008 and in full conformity with the ICH guidelines for Good Clinical Practice (GCP). Written informed consent was given by all volunteers and blood donors.

## Decision letter and Author response

Decision letter https://doi.org/10.7554/eLife.62800.sa1
Author response https://doi.org/10.7554/eLife.62800.sa2

# Additional files

## Supplementary files

• Supplementary file 1. Demographics of volunteers infected by blood or mosquito challenge; includes genetic and non-genetic variables known to influence human immune variation in vitro. Classification of volunteers by host response after blood challenge is described in the results and methods.

• Supplementary file 2. Three metrics that measure the magnitude of the human immune response to blood challenge. [1] The mean variance of the top 50 most variable protein-coding genes in each volunteer. [2] The Euclidean distance travelled during principal component analysis of the 517-gene superset. [3] The number of differentially expressed genes at diagnosis in each group. Note that uninfected control volunteers set a threshold for baseline variation in gene expression through time; a Mann Whitney test was then used to support the observation that unresponsive volunteers were comparable to uninfected controls. Classification of volunteers by host response after blood challenge is described in the results and methods.

• Supplementary file 3. List of the 2028 differentially expressed genes identified in inflammatory volunteers at diagnosis (compared to pre-infection samples). These data underpin the gene ontology network in *Figure 2a*.

• Supplementary file 4. List of the 217 significantly enriched GO terms identified in inflammatory volunteers at diagnosis. GO terms are arranged into their functional groups and groups are ordered by size. The leading GO term in the top 12 functional groups is highlighted in red. On sheet 2, we highlight the top 20 GO terms as ordered by p value.

• Supplementary file 5. List of the 77 differentially expressed genes identified in suppressor volunteers at diagnosis (compared to pre-infection samples). These data underpin the heatmap in *Figure 2d*.

• Supplementary file 6. List of the 893 genes that were differentially expressed between inflammatory and suppressor hosts at diagnosis. There were no differentially expressed genes between these volunteers prior to infection (adj p<0.05).

• Supplementary file 7. Rlog expression values of the 117-gene superset in whole blood after mosquito challenge. These data underpin the heatmap in *Figure 2—figure supplement 1a*. On sheet two we highlight the volunteers in which EdgeR identified these genes as differentially expressed. And on sheet three we look at the 21 genes that were most upregulated in inflammatory volunteers after blood challenge.

• Supplementary file 8. List of the 44 significantly enriched GO terms identified in the 117-gene superset. GO terms are arranged into their functional groups and groups are ordered by size. On sheet two we highlight the top 20 GO terms as ordered by p value.

• Transparent reporting form

## Data availability

Microarray data have been deposited in NCBI GEO, accession code GSE132050 (blood challenge human microarray dataset). Sequencing data have been deposited in EBI EGA, accession code EGAS00001003766 (mosquito challenge human RNA-seq dataset) and EBI ENA, accession code PRJEB33557 (parasite RNA-seq dataset (blood and mosquito challenge)). Metabolomics data have been deposited in MetaboLights, accession code MTBLS1188 (blood challenge metabolomics dataset). Information on clinical trials are available at ClinicalTrials.gov using accession codes NCT02044198 and NCT02905019.

The following datasets were generated:

| Author(s) | Year | Dataset title | Dataset URL | Database and Identifier |
|---|---|---|---|---|
| Milne K, Ivens A, Draper SJ, Rowe A, | 2020 | Longitudinal profiling of the human immune response to | https://www.ncbi.nlm.nih.gov/geo/query/acc. | NCBI Gene Expression Omnibus, GSE132050 |

| Spence PJ | | | Plasmodium falciparum | cgi?acc=GSE132050 | |
|---|---|---|---|---|---|
| Milne K, Reid AJ, Lotkowska ME, Sankaranarayanan G, Sanders MJ, Hill AV, Berriman M, Rowe A, Spence PJ | | 2020 | Immune variation leads to diverse outcomes in human malaria | https://www.ebi.ac.uk/ega/datasets/EGAD00001005790 | EBI EGA accession code, EGAS00001003766 |
| Milne K, Reid AJ, Lotkowska ME, Sankaranarayanan G, Sanders MJ, Hill AV, Berriman M, Draper SJ, Rowe A, Spence PJ | | 2020 | Diverse outcomes of controlled human malaria infection originate from host-intrinsic immune variation and not var gene switching | https://www.ebi.ac.uk/ena/browser/view/PRJEB33557 | ENA accession code , PRJEB33557 |
| Milne K, O'Toole A, Regnault C, Barrett M, Draper SJ, Spence PJ | | 2020 | Immune variation leads to diverse outcomes in human malaria | https://www.ebi.ac.uk/metabolights/search | MetaboLights accession code, MTBLS1188 |

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
