## [Decision Letter]

**Acceptance summary:**

The authors of this manuscript have examined transcriptomic signatures in controlled human malaria infection (CHMI) with *Plasmodium falciparum*. By including analyses of immune signatures and transcription of the major parasite virulence gene families in a temporal fashion, their study provides novel information regarding the impact of inflammatory responses in shaping the expression of certain virulence genes that are important in determining clinical malaria outcomes. The main finding shows that host immune responses vary substantially, influencing time to onset of symptoms, while parasite factors such as replication rates (parasite density) and the absence of marked temporal changes in the expression of variant surface antigens appear not to play a significant role in the initial course of infection. This study represents a large body of carefully collected interesting data that the authors interpreted in an unbiased and well balanced manner. This topic is of importance and relevance to the field of immunoparasitology and specifically, malaria immunology.

**Decision letter after peer review:**

Thank you for submitting your article "Diverse outcomes of controlled human malaria infection originate from immune variation and not *var* gene switching" for consideration by *eLife*. Your article has been reviewed by three peer reviewers, and the evaluation has been overseen by a Reviewing Editor and Miles Davenport as the Senior Editor. The following individuals involved in review of your submission have agreed to reveal their identity: Lars Hviid (Reviewer #1); Karen Day (Reviewer #2).

The reviewers have discussed the reviews with one another and the Reviewing Editor has drafted this decision to help you prepare a revised submission.

Summary:

In this study, the authors conducted analyses of host and parasite transcription signatures in controlled human malaria infection (CHMI) of naïve subjects with *Plasmodium falciparum* 3D7 (mainly blood stage parasites). Specifically, the approaches included analyses of immune signatures and transcription of major parasite virulence gene families (*var* and *rif*) at various time points before and after very low level infections, which allowed the authors to examine this question in unprecedented detail. The main finding shows that host immune responses vary substantially, influencing time to onset of symptoms, while parasite factors such as replication rates (parasite density) and the absence of marked temporal changes in the expression of variant surface antigens appear not to play a significant role in the initial course of infection. The authors conclude that clinical outcome of CHMI depends on the above-mentioned variation in host immune reactivity rather than transcription of particular *var* or *rif* genes.

Although this is well written manuscript describing a study that represents a large body of carefully collected and interpreted data on a topic that is of importance to the malaria community, the conclusion regarding the absence of marked changes in transcripts of *var* and *rif* genes is at variance with the current thinking about the pathogenesis of *P. falciparum* malaria.

As concerns the current studies, the reviewers are requesting extensive revisions of the manuscript. These revisions are directed at the major aspects of the observations/findings and the authors' data interpretations.

There was a general agreement that the results concerning immune transcripts recapitulated to some extent published data. The data showing Plasmodium gene expression and specifically the absence of any changes in the expression of the *var* and *rif* genes generated quite a bit of discussion amongst the reviewers because these results seem at odds what has been reported for host with little malaria experience. In the absence of more definite evidence than what is presented, it is imperative that the authors reduced the rhetoric and entertain a hypothesis that is much better supported by current scientific evidence (i.e., that antigenic variation is very important, indeed). In general, the reviewers agreed that the authors over-interpret their findings on *var* and *rif* gene transcription, and not least their significance and therefore, please tone down your claims.

It would have been reassuring, if the authors could demonstrate that the parasites used in the CHMI were indeed capable of changing transcription pattern in response to selection for specific adhesion specificities, in particular adhesion specificities that have been associated with development of severe complications (e.g., adhesion to EPCR or ICAM-1). Without such a capacity, the analysis of parasite gene transcription loses its significance. While the data in Peters et al., 2002, support the authors' assumption that the parasites used can indeed switch, direct evidence would be preferable. Because the experiments that would be most informative are not really feasible, the authors must acknowledge this in the Discussion.

The short duration of the experiment and the relatively small number of participants may simply be insufficient to yield the transcriptomic changes one might expect on the basis of the existing literature. While the authors acknowledge these caveats, they need to consider them in their conclusions, as well as the Abstract. Another point to consider is the parasite density. If the density is the trigger to switch *var* gene type, then densities may not be high enough in any host. Parasite density is not adequately explored and neither is the contradiction about parasite density and immune status where the inflammatory group had longer infections.

Because the CHMI model necessitated a strict requirement for drug treatment in early infection, it did not essentially recapitulate the natural *P. falciparum* infection. It is possible, therefore, that in this model there was a critical absence of an important positive feedback loops that would have been activated once inflammation started and which created an environment that may be critical in selection of more virulent parasite types by altering endothelial receptor expression. Again all these are crucial issues to consider, as the present story challenges the existing paradigm of the origins of severe malaria.

1) Please comment on the implications of the shortest infection in immunologically unresponsive participants.

2) Subsection “Systemic inflammation coincides with the onset of clinical symptoms”: Why were only three of eight "inflammatory" volunteers included in the analysis discussed? What was the result when all eight were included?

3) Please include discussion on the role of innate immunity to malaria and the impact of inflammation on the induction of adaptive immunity also should be mentioned.

4) A more expanded information and discussion on parasitological aspects are needed to strengthen the conclusion. Please include the following points in your Discussion/conclusion: parasite multiplication which was the same in the various groups, while Infections lasted longer and parasite densities were higher in the inflammatory group. There appeared to be highly variable density thresholds to induce inflammation, and therefore a possibility exists that there was a hidden or sequestered biomass in some individuals as a consequence of an inflammatory response. Please also comment on the synchronicity of infections in individuals as this will impact the density result.

5) One of the earliest reports, Lavstsen et al., 2005, on *var* gene transcription in experimental *P. falciparum* infections, reached a conclusions that transcription of group A and B *var* genes did increase over early days following CHM. Since these are different conclusions re NF54 parasites from which 3D7 was derived, this point requires a discussion. Bachmann et al., 2019, does not actually study naïve hosts but suggests hosts with existing immunity exert more effect on *var* gene profiles than non-immune hosts.

6) Subsection “Immune decision-making in falciparum malaria”, second paragraph: are these the same 4 volunteers identified earlier in the same paragraph?

7) Discussion, third paragraph: It is hard to envisage a scenario in which CHMI would be allowed to proceed further due to the inherent risks to the volunteer of rapidly-increasing parasite density.

8) Discussion, fifth paragraph: it seems that inflammation was only present at the time of treatment and one life cycle (48 h) before, based on Figure 1 – so it may be that there was not enough time for inflammation-mediated alterations in host receptor expression to feed through into the *var* gene profiles by providing a selective advantage to certain adhesive types.

9) "as part of a custom-design multiplex assay from BioLegend" – does that mean other markers were also measured and were not informative? This would be useful to know given the emphasis on an "inflammatory" response when there is no evidence of classic inflammatory cytokines (TNF, IL1 etc.).

10) Subsection “Analysis of parasite *var* and *rifin* expression”: were the *var* reads measured on each of these three days (9, 10, 11)? The Materials and methods don't seem to state which day the samples from the blood challenge were collected for *var* analysis, although it seems clear from results.

11) Please comment on the relevance of these finding to clinical vaccine and drug trials.

12) Please revise the title of your manuscript to reflect the changes in your revised manuscript

Revisions expected in follow-up work:

Given how challenging this work has been, no additional experiments are being suggested for this study, but repeating similar studies with different parasite lines and eventually in malaria endemic areas would be important to consider for future experiments.

---

## [Author Response]

[…] Although this is well written manuscript describing a study that represents a large body of carefully collected and interpreted data on a topic that is of importance to the malaria community, the conclusion regarding the absence of marked changes in transcripts of var and rif genes is at variance with the current thinking about the pathogenesis of P. falciparum malaria.

The absence of marked changes in transcription of *var* and *rifin* genes was unexpected, and in our view it is the counter-intuitive nature of this result that makes it so exciting. We agree that these data do not challenge the paradigm that a subset of *var* genes are associated with severe malaria; instead we believe these data offer new insights into how group A/DC8 variants may come to dominate severe infections. Specifically, they indicate that group A/DC8 variants do not have an intrinsic growth or survival advantage (a leading hypothesis) but instead may rely upon changes in the host environment to promote their selection. We therefore anticipate that this study will stimulate much discussion and further research on the selection of pathogenic parasite variants in vivo. We also agree that CHMI undoubtedly has limitations, and we had intended to describe these fully in our original manuscript. However, we obviously failed to communicate this clearly, so we have substantially re-written parts of the manuscript in order to more clearly and carefully state our conclusions and explore the caveats relating to our data. These changes are described in detail below.

Revisions for this paper:As concerns the current studies, the reviewers are requesting extensive revisions of the manuscript. These revisions are directed at the major aspects of the observations/findings and the authors' data interpretations.There was a general agreement that the results concerning immune transcripts recapitulated to some extent published data. The data showing Plasmodium gene expression and specifically the absence of any changes in the expression of the var and rif genes generated quite a bit of discussion amongst the reviewers because these results seem at odds what has been reported for host with little malaria experience. In the absence of more definite evidence than what is presented, it is imperative that the authors reduced the rhetoric and entertain a hypothesis that is much better supported by current scientific evidence (i.e., that antigenic variation is very important, indeed). In general, the reviewers agreed that the authors over-interpret their findings on var and rif gene transcription, and not least their significance and therefore, please tone down your claims.

We are sorry that our wording gave the impression that we were questioning the importance of antigenic variation in malaria or the role of *var* genes in severe disease – this was not our intention. To qualify our conclusions and describe them in more accurate and measured tones we have made extensive changes to the text. The specific changes made are as follows:

Title: we have changed the title to “Mapping immune variation and *var* gene switching in naïve hosts infected with *Plasmodium falciparum*”. We hope that this title emphasises the unique nature of this study (the transcriptional analysis of both parasite and host through time) without erroneously giving the impression that we are challenging the role of *var* genes in malaria pathogenesis.

Abstract: We have altered the wording of the Abstract to more carefully communicate our conclusions and to explicitly acknowledge the dominant association of specific parasite variants with severe malaria. We have also made it clear that our data are based exclusively on transcriptional analysis:

“When we tracked temporal changes in parasite VSA expression to ask whether variants associated with severe disease rapidly expand in naïve hosts we found no transcriptional evidence to support this hypothesis. These data indicate that parasite variants that dominate severe malaria do not have an intrinsic growth or survival advantage; instead, they presumably rely upon infection-induced changes in their within-host environment for selection.”

Results: we have modified the text to clearly set out the hypothesis that we are testing: “Current theory predicts that group A and DC8 variants should rapidly and preferentially expand in naive hosts because of an intrinsic growth and/or survival advantage”.

And we have qualified our major conclusions:

“Evidently, there was no preferential expansion of group A or DC8 variants within the time-frame of this study” and “The diverse clinical outcomes observed in this trial cannot therefore be easily explained by variation in parasite gene expression; instead, they must largely originate from host-intrinsic immune variation”.

Discussion: we have added three new paragraphs to discuss the limitations of CHMI and to examine the possible reasons why preferential expansion of group A/DC8 variants was not observed within the time-frame of this study. Importantly, of the myriad possibilities we state our preferred hypothesis: that group A/DC8 variants would expand if CHMI were prolonged beyond 10-days and this would result from enhanced selection (due to changes in the host environment) rather than increased switching rates. We make it clear that further experiments are required.

In the Discussion we:

– summarise the limitations of CHMI;

– discuss the limitations imposed by the use of a single parasite genotype, and the possibility that different parasite isolates may vary in their capacity to switch towards group A or DC8 variants;

– discuss the limitations of small sample size, and the possibility that preferential expansion of group A/DC8 variants might be observed if more volunteers are studied;

– discuss the possibility that parasite density is not high enough in CHMI to trigger *var* gene switching/selection;

– discuss the relatively short duration of CHMI and ask whether the expansion of group A/DC8 variants would be observed in longer infections. We then explore in detail why a longer timeframe is likely to be required and the changes that may trigger expansion of pathogenic variants.

And finally, the summary paragraph at the end of the Discussion has been modified to qualify our conclusions and again emphasise the dominant association of specific parasite variants with severe malaria.

It would have been reassuring, if the authors could demonstrate that the parasites used in the CHMI were indeed capable of changing transcription pattern in response to selection for specific adhesion specificities, in particular adhesion specificities that have been associated with development of severe complications (e.g., adhesion to EPCR or ICAM-1). Without such a capacity, the analysis of parasite gene transcription loses its significance. While the data in Peters et al., 2002, support the authors' assumption that the parasites used can indeed switch, direct evidence would be preferable. Because the experiments that would be most informative are not really feasible, the authors must acknowledge this in the Discussion.

We agree that one possible explanation for our data is that the 3D7 parasites used for blood challenge could have lost their ability to switch towards group A/DC8 *var* genes and that this should be assessed in future trials. In the meantime, we have expanded our Discussion to explore this idea further and to state that additional direct experimental evidence are still required:

“The 3D7 clone was cultured for many years before preparation of the inoculum used for blood challenge (Cheng et al., 1997) and could have acquired mutations that affect its behaviour in vivo (Claessens et al., 2017). […] Overall, we consider it possible, but unlikely, that our data can be explained by a mutation in 3D7 that affects *var* gene switching.”

The short duration of the experiment and the relatively small number of participants may simply be insufficient to yield the transcriptomic changes one might expect on the basis of the existing literature. While the authors acknowledge these caveats, they need to consider them in their conclusions, as well as the Abstract. Another point to consider is the parasite density. If the density is the trigger to switch var gene type, then densities may not be high enough in any host. Parasite density is not adequately explored and neither is the contradiction about parasite density and immune status where the inflammatory group had longer infections.

As outlined above, we have carefully re-worded our Abstract and conclusions (subsection “Parasite variants associated with severe disease do not rapidly expand in naïve hosts”) to take account of the limitations of the study. Furthermore, we now discuss in detail the possibility that the short duration of infection, the small number of volunteers or the parasite density observed in CHMI (Discussion) may explain the absence of *var* gene switching/selection.

Because the CHMI model necessitated a strict requirement for drug treatment in early infection, it did not essentially recapitulate the natural P. falciparum infection. It is possible, therefore, that in this model there was a critical absence of an important positive feedback loops that would have been activated once inflammation started and which created an environment that may be critical in selection of more virulent parasite types by altering endothelial receptor expression. Again all these are crucial issues to consider, as the present story challenges the existing paradigm of the origins of severe malaria.

We would like to re-iterate that it was not our intention to challenge the paradigm that group A/DC8 variants are associated with severe malaria, but instead to raise the question of how these variants come to dominate severe infections. Our data do not support the hypothesis that group A/DC8 variants have an intrinsic growth or survival advantage in naïve hosts; instead, our data indicate that their selection will likely require infection-induced changes in the host environment. We agree that infections were probably terminated too early to observe the effects of these changes on parasite selection and this is discussed in detail in the revised manuscript (Discussion). Furthermore, all of our conclusions have been carefully re-worded so that we don’t erroneously give the impression that we are challenging the role of *var* genes in malaria pathogenesis.

1) Please comment on the implications of the shortest infection in immunologically unresponsive participants.

We think it likely that the unresponsive volunteers would have mounted an immune response if their infections had continued for a longer time period:

“Indeed, it remains a possibility that all naive hosts (including those that initially suppress genes involved in myeloid cell activation) will eventually converge on interferon-stimulated inflammation as the infection progresses. […] This metabolic switch may therefore represent a hardwired mechanism of self-regulation because systemic inflammation is an unavoidable fate in naive hosts.”

2) Subsection “Systemic inflammation coincides with the onset of clinical symptoms”: Why were only three of eight "inflammatory" volunteers included in the analysis discussed? What was the result when all eight were included?

Unfortunately, we were only able to carry out metabolomic analysis on a total of 21 samples. We therefore chose 3 time-points from 7 volunteers, and selected 3 from the inflammatory group (those with the highest CXCL10/IFNγ levels), 2 from the unresponsive group and the 2 suppressor volunteers. This has been clarified in the Materials and methods.

3) Please include discussion on the role of innate immunity to malaria and the impact of inflammation on the induction of adaptive immunity also should be mentioned.

The role of innate immunity and inflammation (and the activation of adaptive immunity) is discussed:

“Notably, only inflammatory volunteers developed hallmark signs and symptoms of malaria (e.g. pyrexia and lymphopenia) providing further support for the idea that inflammation causes the earliest manifestations of disease. […] It therefore appears that we might need to re-assess the role of innate immunity in the first hours and days of the pathogenic blood cycle – rather than serving as an early brake on parasite replication it may be that systemic inflammation primarily functions to switch bone marrow production in favour of myelopoiesis and mobilise the required effector cells to the spleen.”

“The innate response to malaria therefore quickly re-structures and co-localises key strands of the immune system; whether this leads to an effective response that can limit parasite burden and minimise collateral damage will depend upon the outcome of cell-cell circuits in the spleen. Measuring the activation and differentiation of CD4^+^ T cells – the key orchestrators of innate and adaptive immunity – would provide a downstream readout of these critical tissue-specific interactions in human volunteers.”

4) A more expanded information and discussion on parasitological aspects are needed to strengthen the conclusion. Please include the following points in your Discussion/conclusion: parasite multiplication which was the same in the various groups, while Infections lasted longer and parasite densities were higher in the inflammatory group. There appeared to be highly variable density thresholds to induce inflammation, and therefore a possibility exists that there was a hidden or sequestered biomass in some individuals as a consequence of an inflammatory response. Please also comment on the synchronicity of infections in individuals as this will impact the density result.

We have expanded our discussion of the parasitological aspects in the revised manuscript (Discussion) and clearly highlighted the limitations of the model. Furthermore, we have qualified all of our conclusions to take account of these limitations.

The apparent contradiction that parasite multiplication rates are comparable between groups but the course of infection (and parasite density) is increased in inflammatory volunteers almost certainly arises from using microscopy for diagnosis. A positive thick blood smear was the primary end-point for drug treatment and inevitably when circulating parasite densities are around the limit of detection (approx. 5-20,000 parasites per ml) diagnosis becomes fairly stochastic. As one example, inflammatory volunteer 016 was diagnosed with a parasite density of 273,247 parasites per ml but if they had been diagnosed just 12-hours earlier their final parasite density would have been just 3,636 parasites per ml. At both time-points this volunteer was inflammatory and symptomatic. Because of the stochastic nature of microscopy we no longer use this as an endpoint in our clinical trials. Length of infection and parasite density are discussed in the Discussion.

With regards to a sequestered parasite biomass: we tried to measure PfHRP2 levels in plasma (which has been used to assess total parasite burden including sequestered parasites). However, PfHRP2 levels were below the limit of detection in all volunteers. It therefore remains possible that a hidden parasite biomass in some volunteers (but not others) explains the different outcomes of infection. However, in our view a more simple explanation is that there is between-host variation in the parasite density threshold required to trigger inflammation. This is well supported by a large body of evidence that comes from the in vitro study of human immune variation (see Bakker et al., 2018; Brodin et al., 2015; Li et al., 2016; Patin et al., 2018; Piasecka et al., 2018; Ter Horst et al., 2016). Furthermore, whilst we are unable to comment in detail on the synchronicity of these sub-patent infections, we do note that the characteristic saw-tooth pattern of parasite density (by qPCR) was observed in every volunteer in our blood challenge model (Figure 3A). This would suggest that sequestration is occurring in all volunteers.

5) One of the earliest reports, Lavstsen et al., 2005, on var gene transcription in experimental P. falciparum infections, reached a conclusions that transcription of group A and B var genes did increase over early days following CHM. Since these are different conclusions re NF54 parasites from which 3D7 was derived, this point requires a discussion. Bachmann et al., 2019, does not actually study naïve hosts but suggests hosts with existing immunity exert more effect on var gene profiles than non-immune hosts.

The Lavstsen et al., 2005, is now discussed:

“This is in contrast to a previous study that suggested *var* gene expression changes within the first few days of blood-stage infection (Lavstsen et al., 2005). However, in that study parasites were isolated from volunteers and then placed into culture for one month before *var* gene profiling – switching may therefore have occurred in vitro confounding interpretation of these data. Our study, which measured *var* gene expression ex vivo, avoids any bias introduced by culturing parasites but nevertheless has limitations that require careful consideration.”

Regarding reference to Bachmann et al., 2019, we apologise for the mistake – the work using parasite strain 7G8 in CHMI by Bachmann et al. has been presented at meetings, but is not published yet. This has been corrected in the revised manuscript.

6) Subsection “Immune decision-making in falciparum malaria”, second paragraph: are these the same 4 volunteers identified earlier in the same paragraph?

Yes, this is shown in Supplementary file 2.

7) Discussion, third paragraph: It is hard to envisage a scenario in which CHMI would be allowed to proceed further due to the inherent risks to the volunteer of rapidly-increasing parasite density.

We have modified this sentence to make it clear that it would be beneficial to prolong CHMI trials if safe to do so:

“CHMI trials that follow volunteers for one or two more blood cycles would be required to explore the consequences of increased Pfck1 transcription and of prolonged systemic inflammation, and consideration should be given as to whether this can be done safely.”

8) Discussion, fifth paragraph: it seems that inflammation was only present at the time of treatment and one life cycle (48 h) before, based on Figure 1 – so it may be that there was not enough time for inflammation-mediated alterations in host receptor expression to feed through into the var gene profiles by providing a selective advantage to certain adhesive types.

We completely agree and discuss this possibility in depth in the Discussion.

9) "as part of a custom-design multiplex assay from BioLegend" – does that mean other markers were also measured and were not informative? This would be useful to know given the emphasis on an "inflammatory" response when there is no evidence of classic inflammatory cytokines (TNF, IL1 etc).

TNF and IL-1b have short half-lives in vivo and would not be expected to increase in plasma within 1-2 days of the start of an inflammatory response. These cytokines are not the triggers for inflammation (that role almost universally falls onto interferon signaling) but rather the downstream effector molecules released by activated myeloid cells. We can detect increased transcription of TNF and IL-1b in whole blood but no increase in circulating protein within the timeframe of this study. For clarity, we have now included reference to TNF in the Discussion:

“However, circulating levels of plasma TNF had not yet increased at diagnosis and so more time is clearly required for interferon signaling to drive the cytokine cascades necessary to activate endothelial cells.”

10) Subsection “Analysis of parasite var and rifin expression”: were the var reads measured on each of these three days (9, 10, 11)? The Materials and methods don't seem to state which day the samples from the blood challenge were collected for var analysis, although it seems clear from results.

We apologise that we did not make this clear. We have now revised the Materials and methods to state exactly when parasites were collected during each study and the circumstances under which *var* reads were merged.

In the blood challenge study, parasites were collected only at diagnosis:

“Blood challenge: 50ml whole blood was drawn into lithium heparin vacutainers at diagnosis (immediately before drug treatment) and white cells were removed by passing the blood through a leucoflex LXT filter (Macopharma).”

In the mosquito challenge study, parasites were collected on days 9, 10 and 11:

“Mosquito challenge: parasites were isolated from 20ml whole blood drawn on days 9, 10 and 11 post-infection according to the blood challenge protocol, with the following modifications.”

Parasite reads were pooled for each volunteer infected by mosquito bite from the data collected on days 9, 10 and 11. This was to ensure sufficient coverage to map *var* gene expression at the start of the blood cycle in every volunteer: “For every volunteer infected by mosquito bite read counts were pooled from days 9, 10 & 11 post-infection to generate a comprehensive parasite transcriptome 2-3 cycles after liver egress.”

11) Please comment on the relevance of these finding to clinical vaccine and drug trials.

A major aim of this study was to map the variation between hosts in their response to infection. Between-host variation has been well documented in vaccine trials (especially for non-malaria infectious diseases) and our data therefore support the need to account for individual variation when calculating sample sizes for vaccine (and drug) trials. Having said that, the sample size in our study is likely not sufficient to reveal the full breadth of immune variation in falciparum malaria and we would therefore be reluctant to include a statement in the manuscript about sample sizes in vaccine or drug trials, as we would not be able to substantiate any claims.

12) Please revise the title of your manuscript to reflect the changes in your revised manuscript.

The title has been changed to “Mapping immune variation and *var* gene switching in naïve hosts infected with *Plasmodium falciparum*”. We hope that this title emphasises the unique nature of this study – the transcriptional analysis of both parasite and host through time.

Revisions expected in follow-up work:Given how challenging this work has been, no additional experiments are being suggested for this study, but repeating similar studies with different parasite lines and eventually in malaria endemic areas would be important to consider for future experiments.

We agree, and have emphasised the need for further research (including different parasite lines and in endemic settings) in our revised Discussion.